# ON THE ERGODIC CONVERGENCE PROPERTIES OF THE PEACEMAN-RACHFORD METHOD AND THEIR APPLICATIONS IN SOLVING LINEAR PROGRAMMING

## ABSTRACT

In this paper, we study the ergodic convergence properties of the Peaceman-Rachford (PR) method with semi-proximal terms for solving convex optimization problems (COPs). By reformulating the PR method as a degenerate proximal point method, for the first time we establish the global convergence of the ergodic sequence generated by the PR method with broadly chosen semi-proximal terms under the assumption that there exists a Karush–Kuhn–Tucker (KKT) solution to the COPs. This result represents a significant departure from previous studies on the non-ergodic convergence of the PR method, which typically requires strong convexity (or strong monotonicity in the reformulated operator) conditions that are hardly satisfied for COPs. Moreover, we establish an ergodic iteration complexity of $O(1/k)$ of the PR method with semi-proximal terms, measured by the objective error, the feasibility violation, and the KKT residual using the $\varepsilon$-subdifferential. Based on these convergence properties, we introduce the solver EPR-LP, using the ergodic sequence of the PR method with semi-proximal terms for solving linear programming (LP) problems. EPR-LP incorporates an adaptive restart strategy and dynamic penalty parameter updates for efficiency and robustness. Extensive numerical experiments on LP benchmark datasets, executed on a high-performance GPU, show that our Julia-based solver outperforms the award-winning solver PDLP at a tolerance level of $10^{-8}$.

## 1 INTRODUCTION

In this paper, we focus on solving the following linear programming (LP) problem:

$$\min_{x \in \mathbb{R}^n} \quad \langle c, x \rangle$$
$$\text{s.t. } A_1 x = b_1$$
$$A_2 x \geq b_2 \tag{1}$$
$$x \in C,$$

where $A_1 \in \mathbb{R}^{m_1 \times n}$, $A_2 \in \mathbb{R}^{m_2 \times n}$, $b_1 \in \mathbb{R}^{m_1}$, $b_2 \in \mathbb{R}^{m_2}$, and $c \in \mathbb{R}^n$. The set $C$ is defined as $C := \{x \in \mathbb{R}^n \mid l \leq x \leq u\}$, with the vectors $l \in (\mathbb{R} \cup \{-\infty\})^n$ and $u \in (\mathbb{R} \cup \{+\infty\})^n$. Let $A = [A_1; A_2] \in \mathbb{R}^{m \times n}$ with $m = m_1 + m_2$ and $b = [b_1; b_2] \in \mathbb{R}^m$. We assume that $A$ is a non-zero matrix, which is also occasionally treated as a linear operator. Then, the dual of problem (1) can be expressed as:

$$\min_{y \in \mathbb{R}^m, z \in \mathbb{R}^n} - \langle b, y \rangle + \delta_D(y) + \delta_C^*(-z)$$
$$\text{s.t. } A^* y + z = c, \tag{2}$$

where $\delta_D(\cdot)$ is the indicator function over $D := \{y = (y_1, y_2) \in \mathbb{R}^{m_1} \times \mathbb{R}_+^{m_2}\}$ and $\delta_C^*(\cdot)$ is the conjugate of $\delta_C(\cdot)$. LP is a fundamental optimization subject in applied mathematics, operations research, and computer science, with a wide range of applications. The interior point methods and the simplex methods are the standard algorithms used in commercial LP solvers such as Gurobi (Gurobi Optimization, LLC, 2024) and CPLEX (IBM, 1987), efficiently solving problems with hundreds to millions of variables and constraints. However, their performance can be inadequate for

problems with huge dimensions (Lu, 2024). Furthermore, these methods are difficult to parallelize, limiting their ability to leverage modern GPUs effectively.

Recently, the first-order methods, especially those based on the alternating direction method of multipliers (ADMM) (O'donoghue et al., 2016; Stellato et al., 2020; Applegate et al., 2021; Lin et al., 2021; O'Donoghue, 2021; Applegate et al., 2023; Deng et al., 2024; Lu & Yang, 2024; Chen et al., 2024) have attracted increasing attention for solving large-scale LP problems, due to their low iteration cost and ease of parallelization. Specifically, let $\sigma > 0$ be a given penalty parameter. The augmented Lagrangian function associated with the dual problem (2), for any $(y, z, x) \in \mathbb{R}^m \times \mathbb{R}^n \times \mathbb{R}^n$, is defined as

$$L_\sigma^{\mathrm{LP}}(y, z; x) := -\langle b, y \rangle + \delta_D(y) + \delta_C^*(-z) + \langle x, A^*y + z - c \rangle + \frac{\sigma}{2}\|A^*y + z - c\|^2.$$

A preconditioned (semi-proximal) ADMM (pADMM) (Xiao et al., 2018) for solving LP is then outlined in Algorithm 1. Consider the case where $\mathcal{S}_1 = 0$. When $\rho = 1$, Algorithm 1 reduces to the Douglas-Rachford method (Gabay, 1983). When $\rho = 2$, it becomes equivalent to the generalized ADMM (GADMM) induced by the Peaceman-Rachford (PR) method (Eckstein & Bertsekas, 1992; Lions & Mercier, 1979). For a more detailed comparison of Algorithm 1 with other algorithms, refer to Xiao et al. (2018) and Sun et al. (2024).

---

**Algorithm 1** A pADMM method for the LP problem (2)

---

1: **Input:** Set the penalty parameter $\sigma > 0$. Choose $\rho \in (0, 2]$. Let $\mathcal{S}_1 : \mathbb{R}^m \to \mathbb{R}^m$ be a self-adjoint positive semidefinite linear operator such that $\mathcal{S}_1 + AA^*$ is positive definite. Choose an initial point $w^0 = (y^0, z^0, x^0) \in D \times \mathrm{dom}(\delta_C^*(-\cdot)) \times \mathbb{R}^n$.

2: **for** $k = 0, 1, ...,$ **do**

3:    Step 1. $\bar{z}^k = \arg\min_{z \in \mathbb{R}^n} \left\{ L_\sigma^{\mathrm{LP}}\left(y^k, z; x^k\right) \right\}$;

4:    Step 2. $\bar{x}^k = x^k + \sigma(A^*y^k + \bar{z}^k - c)$;

5:    Step 3. $\bar{y}^k = \arg\min_{y \in \mathbb{R}^m} \left\{ L_\sigma^{\mathrm{LP}}\left(y, \bar{z}^k; \bar{x}^k\right) + \frac{\sigma}{2}\|y - y^k\|_{\mathcal{S}_1}^2 \right\}$;

6:    Step 4. $w^{k+1} = (1 - \rho)w^k + \rho\bar{w}^k$;

---

Notably, Applegate et al. (Applegate et al., 2021; 2023) developed the award-winning solver PDLP[1], which uses a modified version of the primal-dual hybrid gradient (PDHG) method (Zhu & Chan, 2008) as its base algorithm. This modified PDHG is a special case of Algorithm 1 with $\rho = 1$ (Esser et al., 2010; Chambolle & Pock, 2011). To enhance the modified PDHG's performance, PDLP incorporates several effective techniques, including using the ergodic iterate as a restart point, employing an adaptive update rule for the penalty parameter $\sigma$, and implementing a line search strategy. It is worth noting that, compared to non-ergodic sequences, the ergodic sequence of the semi-proximal ADMM (sPADMM) (Fazel et al., 2013), including the modified PDHG (Esser et al., 2010; Chambolle & Pock, 2011), achieves a better $O(1/k)$ iteration complexity with respect to the objective error and the feasibility violation (Cui et al., 2016). Consequently, the GPU implementation of PDLP (cuPDLP.jl (Lu & Yang, 2023) and cuPDLP-c (Lu et al., 2023)) has demonstrated advantages over commercial LP solvers like Gurobi (Gurobi Optimization, LLC, 2024) and COPT (Ge et al., 2022), particularly for large-scale LP problems.

More recently, Chen et al. (Chen et al., 2024) introduced HPR-LP, an implementation of the Halpern Peaceman-Rachford (HPR) method (Sun et al., 2024) with semi-proximal terms for solving LP problems. This method integrates the Halpern iteration (Halpern, 1967; Lieder, 2021) into the PR method (Lions & Mercier, 1979; Eckstein & Bertsekas, 1992) with semi-proximal terms, corresponding to Algorithm 1 using $\rho = 2$. It achieves an iteration complexity of $O(1/k)$ for the Karush–Kuhn–Tucker (KKT) residual and the objective error. The Julia implementation of HPR-LP has demonstrated superior performance compared to PDLP on classical LP benchmark datasets.

A key difference between HPR-LP and PDLP is the choice of $\rho$. PDLP uses a conservative $\rho = 1$ and adopts a line-search strategy to enhance performance, while HPR-LP employs a more aggressive

---

[1]The authors of Applegate et al. (2021; 2023) received the Beale–Orchard-Hays Prize for Excellence in Computational Mathematical Programming at the 25th International Symposium on Mathematical Programming (https://ismp2024.gerad.ca/), held from July 21-26, 2024, in Montréal, Canada.

$\rho = 2$. It is well known that the PR method ($\rho = 2$) is faster than the DR method ($\rho = 1$) when the PR method converges (Lions & Mercier, 1979). This difference in $\rho$ selection motivates our study of using the ergodic sequence of the PR method with semi-proximal terms to solve LP problems.

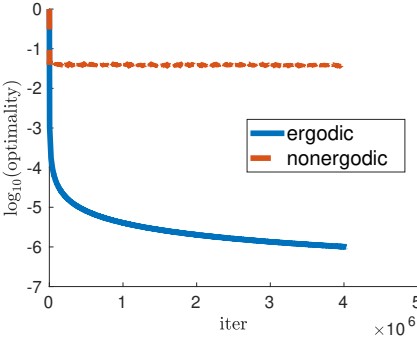

Figure 1: The performance of PR, using ergodic and non-ergodic sequences, in solving the LP instance "datt256" from Mittelmann's LP benchmark set. The optimality is evaluated based on the relative primal-dual infeasibility and the relative duality gap.

To the best of our knowledge, the ergodic convergence of the PR method with semi-proximal terms (corresponding to Algorithm 1 with $\rho = 2$) is still unknown. In this paper, we address this gap by proving, for the first time, the global convergence of the ergodic sequence using the theory of the degenerate proximal point method (dPPM) (Bredies et al., 2022). This result marks a significant departure from previous studies on the non-ergodic convergence of the PR method, which typically relies on strong convexity that is not met in LP (see Figure 1). Moreover, we also investigate the ergodic iteration complexity of the pADMM in Algorithm 1 in terms of the objective error, the feasibility violation, and the KKT residual based on $\varepsilon$-subdifferential (Rockafellar, 1970). The main contributions of this paper can be summarized as follows:

- For a more general convex optimization problem (COP), by reformulating the pADMM with $\rho \in (0, 2]$ as a dPPM, we establish the ergodic convergence of the dPPM, thereby proving the ergodic convergence of the pADMM through this reformulation.
- For solving COP, we establish the ergodic iteration complexity of $O(1/k)$ for the pADMM with $\rho \in (0, 2]$ in terms of the objective error, the feasibility violation, and the KKT residual based on $\varepsilon$-subdifferential.
- We propose the solver EPR-LP, an implementation of the **e**rgodic sequence of **PR** method (corresponding to Algorithm 1 with $\rho = 2$) to solve **LP** problems, incorporating an adaptive restart strategy and dynamic penalty parameter updates. Extensive numerical experiments on LP benchmark datasets using a GPU demonstrate the superior performance of the Julia implementation of EPR-LP compared to PDLP (Lu & Yang, 2023).

The remainder of this paper is structured as follows: Section 2 establishes the ergodic convergence and iteration complexity of pADMM for solving convex optimization problems. The detailed implementation of EPR-LP is discussed in Section 3. Section 4 presents the results of numerical experiments on LP benchmark datasets. Finally, we conclude the paper in Section 5.

**Notation.** Let $\mathbb{U}$, $\mathbb{W}$, $\mathbb{X}$, $\mathbb{Y}$, and $\mathbb{Z}$ be finite-dimensional real Euclidean spaces, each equipped with an inner product $\langle \cdot, \cdot \rangle$ and its corresponding norm $\| \cdot \|$. For any convex function $f : \mathbb{X} \to (-\infty, +\infty]$, we define its effective domain as $\mathrm{dom}(f) := \{x \in \mathbb{X} : f(x) < \infty\}$, its conjugate as $f^*(z) := \sup_{x \in \mathbb{X}} \{\langle x, z \rangle - f(x)\}$, $z \in \mathbb{X}$, and its proximal mapping as $\mathrm{Prox}_f(x) := \arg\min_{z \in \mathbb{X}} \{f(z) + \frac{1}{2}\|z - x\|^2\}$, $x \in \mathbb{X}$. Furthermore, consider a closed convex set $C \subseteq \mathbb{X}$. We define the distance from $x \in \mathbb{X}$ to $C$ as $\mathrm{dist}(x, C) := \inf_{z \in C} \|z - x\|$, and we express the Euclidean projection of $x$ onto $C$ as $\Pi_C(x) := \arg\min\{\|x - z\| \mid z \in C\}$. Moreover, for a linear operator $A : \mathbb{X} \to \mathbb{Y}$, we denote its adjoint by $A^*$ and express its spectral norm as $\|A\| := \sup_{\|x\| \leq 1} \|Ax\|$. Finally, for any self-adjoint, positive semidefinite linear operator $\mathcal{M} : \mathbb{X} \to \mathbb{X}$, we define the semi-norm as $\|x\|_{\mathcal{M}} := \sqrt{\langle x, \mathcal{M}x \rangle}$ for any $x \in \mathbb{X}$.

## 2 ERGODIC CONVERGENCE PROPERTIES OF THE PADMM

In this section, we study the ergodic convergence properties of the pADMM method, including the PR method, for solving COP. We begin by reformulating the pADMM method as a dPPM. Next, we prove the ergodic convergence of the dPPM, thereby establishing the ergodic convergence of pADMM. Finally, we analyze the ergodic iteration complexity of the pADMM method.

### 2.1 ERGODIC CONVERGENCE OF PADMM

Consider the following COP:

$$\min_{y \in \mathbb{Y}, z \in \mathbb{Z}} \quad f_1(y) + f_2(z) \tag{3}$$
$$\text{s.t.} \quad B_1 y + B_2 z = c,$$

where $f_1 : \mathbb{Y} \to (-\infty, +\infty]$ and $f_2 : \mathbb{Z} \to (-\infty, +\infty]$ are proper closed convex functions, $B_1 : \mathbb{Y} \to \mathbb{X}$ and $B_2 : \mathbb{Z} \to \mathbb{X}$ are given linear operators, and $c \in \mathbb{X}$ is a given vector. Given a penalty parameter $\sigma > 0$, the augmented Lagrangian function for problem (3) is defined, for any $(y, z, x) \in \mathbb{Y} \times \mathbb{Z} \times \mathbb{X}$, as

$$L_\sigma(y, z; x) := f_1(y) + f_2(z) + \langle x, B_1 y + B_2 z - c \rangle + \frac{\sigma}{2} \|B_1 y + B_2 z - c\|^2.$$

The dual of problem (3) is given by

$$\max_{x \in \mathbb{X}} \left\{ -f_1^*(-B_1^* x) - f_2^*(-B_2^* x) - \langle c, x \rangle \right\}. \tag{4}$$

Let $w := (y, z, x) \in \mathbb{W} := \mathbb{Y} \times \mathbb{Z} \times \mathbb{X}$. The pADMM method (Xiao et al., 2018) for solving problem (3) is outlined in Algorithm 2.

---

**Algorithm 2** A pADMM for solving COP (3)

---

1: Input: Let $\mathcal{T}_1$ and $\mathcal{T}_2$ be two self-adjoint, positive semidefinite operators on $\mathbb{Y}$ and $\mathbb{Z}$, respectively. Select an initial point $w^0 = (y^0, z^0, x^0) \in \text{dom}(f_1) \times \text{dom}(f_2) \times \mathbb{X}$. Set the parameters $\sigma > 0$ and $\rho \in (0, 2]$.
2: **for** $k = 0, 1, ...,$ **do**
3:     Step 1. $\bar{z}^k = \arg\min_{z \in \mathbb{Z}} \left\{ L_\sigma(y^k, z; x^k) + \frac{1}{2}\|z - z^k\|_{\mathcal{T}_2}^2 \right\}$;
4:     Step 2. $\bar{x}^k = x^k + \sigma(B_1 y^k + B_2 \bar{z}^k - c)$;
5:     Step 3. $\bar{y}^k = \arg\min_{y \in \mathbb{Y}} \left\{ L_\sigma(y, \bar{z}^k; \bar{x}^k) + \frac{1}{2}\|y - y^k\|_{\mathcal{T}_1}^2 \right\}$;
6:     Step 4. $w^{k+1} = (1 - \rho)w^k + \rho \bar{w}^k$;

---

As shown in Rockafellar (1970, Corollary 28.3.1), a pair $(y^*, z^*) \in \mathbb{Y} \times \mathbb{Z}$ is an optimal solution to problem (3) if and only if there exists $x^* \in \mathbb{X}$ such that $(y^*, z^*, x^*)$ satisfies the following KKT system:

$$-B_1^* x^* \in \partial f_1(y^*), \quad -B_2^* x^* \in \partial f_2(z^*), \quad B_1 y^* + B_2 z^* - c = 0, \tag{5}$$

where $\partial f_i$ denotes the subdifferential mapping of $f_i$ for $i = 1, 2$. To discuss the ergodic convergence of pADMM as outlined in Algorithm 2, we make the following assumption:

**Assumption 2.1.** *The KKT system* (5) *has a nonempty solution set.*

Under Assumption 2.1, solving problem (3) is equivalent to finding $w \in \mathbb{W}$ such that $0 \in \mathcal{T}w$, where the maximal monotone operator $\mathcal{T}$ is defined as

$$\mathcal{T}w = \begin{pmatrix} \partial f_1(y) + B_1^* x \\ \partial f_2(z) + B_2^* x \\ c - B_1 y - B_2 z \end{pmatrix}, \quad \forall w = (y, z, x) \in \mathbb{W}. \tag{6}$$

On the other hand, since $f_1$ and $f_2$ are proper closed convex functions, there exist two self-adjoint, positive semidefinite operators, $\Sigma_{f_1}$ and $\Sigma_{f_2}$, such that for all $y, \hat{y} \in \text{dom}(f_1)$, $\phi \in \partial f_1(y)$, and $\hat{\phi} \in \partial f_1(\hat{y})$, the following hold:

$$f_1(y) \geq f_1(\hat{y}) + \langle \hat{\phi}, y - \hat{y} \rangle + \frac{1}{2}\|y - \hat{y}\|_{\Sigma_{f_1}}^2, \quad \text{and} \quad \langle \phi - \hat{\phi}, y - \hat{y} \rangle \geq \|y - \hat{y}\|_{\Sigma_{f_1}}^2,$$

and for all $z, \hat{z} \in \mathrm{dom}(f_2)$, $\varphi \in \partial f_2(z)$, and $\hat{\varphi} \in \partial f_2(\hat{z})$:

$$f_2(z) \geq f_2(\hat{z}) + \langle \hat{\varphi}, z - \hat{z} \rangle + \frac{1}{2}\|z - \hat{z}\|_{\Sigma_{f_2}}^2, \quad \text{and} \quad \langle \varphi - \hat{\varphi}, z - \hat{z} \rangle \geq \|z - \hat{z}\|_{\Sigma_{f_2}}^2.$$

To ensure that each step of the pADMM is well-defined, we also make the following assumption:

**Assumption 2.2.** *Both $\Sigma_{f_1} + B_1^* B_1 + \mathcal{T}_1$ and $\Sigma_{f_2} + B_2^* B_2 + \mathcal{T}_2$ are positive definite.*

Define the self-adjoint linear operator $\mathcal{M} : \mathbb{W} \to \mathbb{W}$ as follows:

$$\mathcal{M} = \begin{bmatrix} \sigma B_1^* B_1 + \mathcal{T}_1 & 0 & B_1^* \\ 0 & \mathcal{T}_2 & 0 \\ B_1 & 0 & \sigma^{-1}\mathcal{I} \end{bmatrix}. \tag{7}$$

According to Sun et al. (2024, Proposition 3.2), we have the following equivalence between Algorithm 2 and the dPPM (Bredies et al., 2022).

**Proposition 2.1.** *Suppose that Assumption 2.2 holds. Consider the operators $\mathcal{T}$ defined in (6) and $\mathcal{M}$ defined in (7). Then the sequence $\{w^k\}$ generated by the pADMM in Algorithm 2 coincides with the sequence $\{w^k\}$ generated by the dPPM as follows:*

$$\bar{w}^k = \widehat{\mathcal{T}}w^k = (\mathcal{M} + \mathcal{T})^{-1}\mathcal{M}w^k, \quad w^{k+1} = (1 - \rho)w^k + \rho\bar{w}^k \tag{8}$$

*with the same initial point $w^0 \in \mathbb{W}$. Additionally, $\mathcal{M}$ is an admissible preconditioner[2] such that $(\mathcal{M} + \mathcal{T})^{-1}$ is Lipschitz continuous.*

We now analyze the ergodic convergence of the dPPM for a general maximal monotone operator $\mathcal{T}$ and an admissible preconditioner $\mathcal{M}$, encompassing Algorithm 2 for solving the COP problem (3). To this end, we define the following two ergodic sequences:

$$w_a^k := \frac{1}{k+1}\sum_{t=0}^k w^t, \quad \bar{w}_a^k := \frac{1}{k+1}\sum_{t=0}^k \bar{w}^t, \quad \forall k \geq 0, \tag{9}$$

where the sequences $\{w^t\}$ and $\{\bar{w}^t\}$ are generated by the dPPM in (8). Note that, for a maximal monotone operator $\mathcal{T} : \mathbb{W} \to 2^{\mathbb{W}}$ and $\varepsilon \geq 0$, the $\varepsilon$-enlargement of $\mathcal{T}$ at $w$ (Burachik et al., 1997) is defined as

$$\mathcal{T}^\varepsilon(w) = \{v \in \mathbb{W} : \langle w - w', v - v' \rangle \geq -\varepsilon, \forall (w', v') \in \mathrm{gph}(\mathcal{T})\},$$

where $\mathrm{gph}(\mathcal{T}) = \{(w, v) \in \mathbb{W} \times \mathbb{W} \mid v \in \mathcal{T}w\}$. Using the $\varepsilon$-enlargement of $\mathcal{T}$, we can derive the following proposition regarding the ergodic convergence properties of the dPPM.

**Proposition 2.2.** *Let $\mathcal{T} : \mathbb{W} \to 2^{\mathbb{W}}$ be a maximal monotone operator with $\mathcal{T}^{-1}(0) \neq \emptyset$, and let $\mathcal{M}$ be an admissible preconditioner. Then the ergodic sequences $\{\bar{w}_a^k\}$ and $\{w_a^k\}$, generated by the dPPM in (8) with $\rho \in (0, 2]$, satisfy the following properties for any $k \geq 0$:*

*(a) $\|\bar{w}_a^k - w_a^k\|_{\mathcal{M}} \leq \frac{2}{\rho(k+1)}\|w^0 - w^*\|_{\mathcal{M}}, \quad \forall w^* \in \mathcal{T}^{-1}(0);$*

*(b) $\mathcal{M}(w_a^k - \bar{w}_a^k) \in \mathcal{T}^{\bar{\varepsilon}_a^k}(\bar{w}_a^k)$, where $\bar{\varepsilon}_a^k := \frac{1}{k+1}\sum_{t=0}^k \langle \bar{w}^t - \bar{w}_a^k, w^t - \bar{w}^t \rangle_{\mathcal{M}}$ and*

$$0 \leq \bar{\varepsilon}_a^k \leq \frac{1}{2\rho(k+1)}\|w^0 - w^*\|_{\mathcal{M}}^2, \quad \forall w^* \in \mathcal{T}^{-1}(0).$$

*Remark 2.1. Monteiro & Sim (2018) used a non-Euclidean hybrid proximal extragradient framework to obtain a similar result to Proposition 2.2, which could be adapted to analyze the ergodic iteration complexity of Algorithm 2 with $\mathcal{T}_1 = 0$, $\mathcal{T}_2 = 0$, and $\rho = 2$. In contrast, Proposition 2.2, developed using the dPPM framework, is more general as it only requires $\mathcal{M}$ to be positive semidefinite. This broader setting allows for the analysis of the ergodic iteration complexity of Algorithm 2 for positive semidefinite linear operators $\mathcal{T}_1$ and $\mathcal{T}_2$.*

---

[2]In Bredies et al. (2022), an admissible preconditioner for the operator $\mathcal{T} : \mathbb{W} \to 2^{\mathbb{W}}$ is a linear, bounded, self-adjoint, and positive semidefinite operator $\mathcal{M} : \mathbb{W} \to \mathbb{W}$ such that $\widehat{\mathcal{T}} = (\mathcal{M} + \mathcal{T})^{-1}\mathcal{M}$ is single-valued and has full domain.

**Theorem 2.1.** *Let $\mathcal{T} : \mathbb{W} \to 2^{\mathbb{W}}$ be a maximal monotone operator with $\mathcal{T}^{-1}(0) \neq \emptyset$, and let $\mathcal{M}$ be an admissible preconditioner such that $(\mathcal{M} + \mathcal{T})^{-1}$ is $L$-Lipschitz continuous. Then, the ergodic sequence $\{\bar{w}_a^k\}$ generated by the dPPM in (8) with $\rho \in (0, 2]$ converges to a point in $\mathcal{T}^{-1}(0)$.*

**Remark 2.2.** *Note that we only assume $\mathcal{M}$ to be positive semidefinite in Theorem 2.1. If $\mathcal{M}$ is positive definite, one can directly apply Baillon's nonlinear ergodic theorem (Baillon, 1975) to obtain the ergodic convergence.*

Consider the ergodic sequence of the pADMM:

$$(y_a^k, z_a^k, x_a^k) := \frac{1}{k+1} \sum_{t=0}^k (y^t, z^t, x^t), \quad (\bar{y}_a^k, \bar{z}_a^k, \bar{x}_a^k) := \frac{1}{k+1} \sum_{t=0}^k (\bar{y}^t, \bar{z}^t, \bar{x}^t), \quad k \geq 0,$$

where the sequences $\{w^t\} = \{(y^t, z^t, x^t)\}$ and $\{\bar{w}^t\} = \{(\bar{y}^t, \bar{z}^t, \bar{x}^t)\}$ are generated by the pADMM in Algorithm 2. The equivalence established in Proposition 2.1 demonstrates that the ergodic convergence of the dPPM in Theorem 2.1 can be leveraged to derive the ergodic convergence of the pADMM, as stated in the following corollary.

**Corollary 2.1.** *Suppose that Assumptions 2.1 and 2.2 hold. Then the ergodic sequence $\{\bar{w}_a^k\} = \{(\bar{y}_a^k, \bar{z}_a^k, \bar{x}_a^k)\}$, generated by the pADMM in Algorithm 2, converges to the point $w^* = (y^*, z^*, x^*)$, where $(y^*, z^*)$ is a solution to problem (3), and $x^*$ is a solution to problem (4).*

## 2.2 Ergodic iteration complexity of pADMM

We introduce the concept of the $\varepsilon$-subgradient of a convex function $f$ (Rockafellar, 1970):

**Definition 2.1.** *Let $f : \mathbb{X} \to (-\infty, +\infty]$ be a proper convex function, and let $\bar{x} \in \mathrm{dom}(f)$. Given $\varepsilon \geq 0$, the $\varepsilon$-subgradient of $f$ at $\bar{x}$ is defined as*

$$\partial_\varepsilon f(\bar{x}) := \{x^* \in \mathbb{X}^* \mid \langle x^*, x - \bar{x} \rangle \leq f(x) - f(\bar{x}) + \varepsilon, \, \forall x \in \mathbb{X}\}.$$

Based on the optimality conditions of each subproblem in the pADMM, we derive the following lemma regarding the ergodic sequence $\{\bar{w}_a^k\}$ using the $\varepsilon$-subgradient.

**Lemma 2.1.** *Suppose that Assumptions 2.1 and 2.2 hold. Let $\{(\bar{y}_a^k, \bar{z}_a^k, \bar{x}_a^k)\}$ be the sequence generated by Algorithm 2 and Let $w^* = (y^*, z^*, x^*)$ be a solution to the KKT system (5). The following things hold: for any $k \geq 0$,*

$$\begin{cases} -B_2^* \bar{x}_a^k - \mathcal{T}_2(\bar{z}_a^k - z_a^k) \in \partial_{\bar{\varepsilon}_z^k} f_2(\bar{z}_a^k), \\ -B_1^*(\bar{x}_a^k + \sigma(B_1 \bar{y}_a^k + B_2 \bar{z}_a^k - c)) - \mathcal{T}_1(\bar{y}_a^k - y_a^k) \in \partial_{\bar{\varepsilon}_y^k} f_1(\bar{y}_a^k), \end{cases} \tag{10}$$

*where*

$$\begin{cases} \bar{\varepsilon}_z^k = \frac{1}{k+1} \sum_{t=0}^k \langle -B_2^* \bar{x}^t - \mathcal{T}_2(\bar{z}^t - z^t), \bar{z}^t - \bar{z}_a^k \rangle \geq 0, \\ \bar{\varepsilon}_y^k = \frac{1}{k+1} \sum_{t=0}^k \langle -B_1^*(\bar{x}^t + \sigma(B_1 \bar{y}^t + B_2 \bar{z}^t - c)) - \mathcal{T}_1(\bar{y}^t - y^t), \bar{y}^t - \bar{y}_a^k \rangle \geq 0, \end{cases} \tag{11}$$

*and*

$$\bar{\varepsilon}_z^k + \bar{\varepsilon}_y^k \leq \frac{1}{2\rho(k+1)} \|w^0 - w^*\|_{\mathcal{M}}^2. \tag{12}$$

Furthermore, to estimate the objective error, we define

$$h(\bar{y}_a^k, \bar{z}_a^k) := f_1(\bar{y}_a^k) + f_2(\bar{z}_a^k) - f_1(y^*) - f_2(z^*), \quad \forall k \geq 0,$$

where $(y^*, z^*)$ is the limit point of the sequence $\{(\bar{y}_a^k, \bar{z}_a^k)\}$. Then the ergodic iteration complexity of pADMM is established in Theorem 2.2.

**Theorem 2.2.** *Suppose that Assumptions 2.1 and 2.2 hold. Let $\{(\bar{y}_a^k, \bar{z}_a^k, \bar{x}_a^k)\}$ be the ergodic sequence generated by Algorithm 2 with $\rho \in (0, 2]$. Let $w^* = (y^*, z^*, x^*)$ be a solution to the KKT system (5), and $R_0 = \|w^0 - w^*\|_{\mathcal{M}}$. For all $k \geq 0$, the following bound holds:*

$$\mathrm{dist}\left(0, \partial_{\bar{\varepsilon}_y^k} f_1(\bar{y}_a^k) + B_1^* \bar{x}_a^k\right) + \mathrm{dist}\left(0, \partial_{\bar{\varepsilon}_z^k} f_2(\bar{z}_a^k) + B_2^* \bar{x}_a^k\right) + \|B_1 \bar{y}_a^k + B_2 \bar{z}_a^k - c\|$$

$$\leq \left(\frac{\sigma\|B_1^*\|+1}{\sqrt{\sigma}} + \|\sqrt{\mathcal{T}_2}\| + \|\sqrt{\mathcal{T}_1}\|\right) \frac{2R_0}{\rho(k+1)}, \tag{13}$$

*where $\bar{\varepsilon}_z^k + \bar{\varepsilon}_y^k \leq \frac{1}{2\rho(k+1)} \|w^0 - w^*\|_{\mathcal{M}}^2$. Moreover,*

$$\left(\frac{-1}{\sqrt{\sigma}} \|x^*\|\right) \frac{2R_0}{\rho(k+1)} \leq h(\bar{y}_a^k, \bar{z}_a^k) \leq (R_0 + 4\sqrt{\sigma} \|B_1 y^*\|) \frac{R_0}{2\rho(k+1)} + \frac{\|x^0 + \sigma B_1 y^0\|^2}{2\rho(k+1)}. \tag{14}$$

**Remark 2.3.** *For the ergodic iteration complexity about LP, please refer to Appendix B. The pADMM with $\rho \in (0, 2)$ achieves only a non-ergodic iteration complexity of $o(1/\sqrt{k})$ with respect to the objective error, feasibility violation, and KKT residual, as established in Appendix C. A detailed comparison of the iteration complexities of related algorithms can be found in Appendix D.*

Based on the ergodic complexity results in Theorem 2.2, the optimal choice for $\rho$ is 2, resulting in an ergodic PR (EPR) method with semi-proximal terms. In the next section, we apply this EPR method to solve large-scale LP problems.

## 3 A PEACEMAN-RACHFORD METHOD USING ERGODIC SEQUENCE FOR SOLVING LP

In this section, we introduce the solver EPR-LP for large-scale LP problems (Algorithm 3), which incorporates a restart strategy and adaptive updates of the penalty parameter $\sigma$ into the EPR method with semi-proximal terms.

---

**Algorithm 3** EPR-LP: A Peaceman-Rachford method using ergodic sequence for the problem (2)

---

1: **Input:** Let $\mathcal{S}_1 : \mathbb{R}^m \to \mathbb{R}^m$ be a self-adjoint, positive semidefinite linear operator such that $\mathcal{S}_1 + AA^*$ is positive definite. Choose an initial point $w^{0,0} = (y^{0,0}, z^{0,0}, x^{0,0}) \in D \times \mathrm{dom}(\delta_C^*(-\cdot)) \times \mathbb{R}^n$.

2: **Initialization**: Set the outer loop counter $r = 0$, the total loop counter $k = 0$, and the initial penalty parameter $\sigma_0 > 0$.

3: **repeat**

4:     initialize the inner loop: set inner loop counter $t = 0$;

5:     **repeat**

6:         $\bar{z}^{r,t} = \underset{z \in \mathbb{R}^n}{\arg\min} \left\{ L_{\sigma_r}^{\mathrm{LP}} \left( y^{r,t}, z; x^{r,t} \right) \right\}$;

7:         $\bar{x}^{r,t} = x^{r,t} + \sigma_r (A^* y^{r,t} + \bar{z}^{r,t} - c)$;

8:         $\bar{y}^{r,t} = \underset{y \in \mathbb{R}^m}{\arg\min} \left\{ L_{\sigma_r}^{\mathrm{LP}} \left( y, \bar{z}^{r,t}; \bar{x}^{r,t} \right) + \frac{\sigma_r}{2} \| y - y^{r,t} \|_{\mathcal{S}_1}^2 \right\}$;

9:         $w^{r,t+1} = 2\bar{w}^{r,t} - w^{r,t}$;

10:         $\bar{w}_a^{r,t} = \sum_{i=0}^{t} \frac{1}{t+1} \bar{w}^{r,i}$;

11:         $t = t + 1, k = k + 1$;

12:     **until** one of the restart criteria holds or termination criteria hold

13:     **restart the inner loop:** $\tau_r = t, w^{r+1,0} = \bar{w}_a^{r,\tau_r}$,

14:     $\sigma_{r+1} = \mathbf{SigmaUpdate}(\bar{w}_a^{r,\tau_r}, w^{r,0}, \mathcal{S}_1, A), r = r + 1$;

15: **until** termination criteria hold

16: **Output:** $\{\bar{w}_a^{r,t}\}$.

---

**Remark 3.1.** *If line 9 in Algorithm 3 is modified to $w^{r,t+1} = \bar{w}^{r,t}$, the resulting method is referred to as EDR-LP, which uses the ergodic sequence of the DR method with semi-proximal terms for solving LP. In the numerical experiments, we compare the performance of EPR-LP and EDR-LP to evaluate the impact of the parameter $\rho$.*

### 3.1 RESTART STRATEGY

The EPR method with semi-proximal terms achieves an ergodic iteration complexity of $O(1/k)$ for the objective error, the feasibility violation, and the KKT condition, as shown in Theorem 2.2. This result, derived from Proposition 2.2, guides the restart criteria using the merit function:

$$R_{r,t} := \| w^{r,t} - w^* \|_{\mathcal{M}}, \quad \forall r \geq 0, \ t \geq 0,$$

where $w^*$ is a solution to the KKT system (5). $R_{r,0}$ gives the upper bound from Proposition 2.2. A natural restart occurs when $R_{r,t} \leq \alpha_1 R_{r,0}$, with $\alpha_1 \in (0, 1)$. In practice, if $\alpha_1$ is too small, the algorithm may fail to reduce sufficiently, so we also consider the inner loop length and oscillation in $R_{r,t}$. These ideas are implemented in PDLP with other merit functions (Applegate et al., 2021; Lu & Yang, 2023). In addition, since $w^*$ is unknown, we replace $R_{r,t}$ with the following weighted primal and dual infeasibility:

$$\widetilde{R}_{r,t} = \sqrt{\sigma_r^{-1}\|\Pi_D(b - A\bar{x}_a^{r,t})\|^2 + \sigma_r\|c - A^*\bar{y}_a^{r,t} - \bar{z}_a^{r,t}\|^2}.$$

Consequently, the restart criteria in EPR-LP are defined as follows:

1. Sufficient decay of $\widetilde{R}_{r,t}$:
$$\widetilde{R}_{r,t} \leq \alpha_1 \widetilde{R}_{r,0}; \tag{15}$$

2. Necessary decay + no local progress of $\widetilde{R}_{r,t}$:
$$\widetilde{R}_{r,t} \leq \alpha_2 \widetilde{R}_{r,0} \quad \text{and} \quad \widetilde{R}_{r,t+1} > \widetilde{R}_{r,t}; \tag{16}$$

3. Long inner loop:
$$t \geq \alpha_3 k, \tag{17}$$

where $\alpha_1 \in (0, \alpha_2)$, $\alpha_2 \in (0, 1)$, and $\alpha_3 \in (0, 1)$. In EPR-LP, we set $\alpha_1 = 0.2$, $\alpha_2 = 0.6$, and $\alpha_3 = 0.2$. Once any of the three restart criteria is met, we restart the inner loop for the $(r + 1)$-th iteration, set $w^{r+1,0} = \bar{w}_a^{r,\tau_r}$, and update $\sigma_{r+1}$.

### 3.2 UPDATE RULE FOR $\sigma$

Motivated by the update rule for $\sigma$ based on the iteration complexity of pADMM proposed in Chen et al. (2024), we update $\sigma_{r+1}$ at the $(r + 1)$-th restart for any $r \geq 0$ by solving the following optimization problem:
$$\sigma_{r+1} := \arg\min_\sigma \left\|w^{r+1,0} - w^*\right\|_{\mathcal{M}}^2, \tag{18}$$

where $\left\|w^{r+1,0} - w^*\right\|_{\mathcal{M}}$ represents the upper bound of the complexity results in Proposition 2.2 at the $(r + 1)$-th outer loop. A smaller upper bound is expected to lead to a smaller residual $\|\bar{w}_a^{r+1,t} - w_a^{r+1,t}\|_{\mathcal{M}}$ for any $t \geq 0$. Specializing $\mathcal{M}$ in (7) to Algorithm 1, we derive the following:

$$\begin{aligned} \sigma_{r+1} &= \arg\min_\sigma \left\|w^{r+1,0} - w^*\right\|_{\mathcal{M}}^2 \\ &= \sqrt{\frac{\|x^{r+1,0} - x^*\|^2}{\|y^{r+1,0} - y^*\|_{\mathcal{S}_1}^2 + \|A^*(y^{r+1,0} - y^*)\|^2}}. \end{aligned} \tag{19}$$

Since computing $\|x^{r+1,0} - x^*\|$ and $\|y^{r+1,0} - y^*\|_{\mathcal{S}_1}^2 + \|A^*(y^{r+1,0} - y^*)\|^2$ is not implementable, we approximate these terms in EPR-LP using:

$$\Delta_x := \|\bar{x}_a^{r,\tau_r} - x^{r,0}\| \quad \text{and} \quad \Delta_y := \sqrt{\|\bar{y}_a^{r,\tau_r} - y^{r,0}\|_{\mathcal{S}_1}^2 + \|A^*(\bar{y}_a^{r,\tau_r} - y^{r,0})\|^2}, \tag{20}$$

respectively. Consequently, we update $\sigma_{r+1}$ as follows:

$$\sigma_{r+1} = \frac{\Delta_x}{\Delta_y}. \tag{21}$$

Because the approximations $\Delta_x$ and $\Delta_y$ may deviate significantly from their true values, we implement some safeguards for the $\sigma$ update rule. For more details, see Appendix E. Furthermore, to ensure that the EPR-LP method has an explicit update formula for $\bar{y}^{r,t}$ for solving general LP problems, we select
$$\mathcal{S}_1 = \lambda I_m - AA^*,$$

with $\lambda \geq \lambda_1(AA^*)$, as proposed in Esser et al. (2010); Chambolle & Pock (2011); Xu & Wu (2011). In this case, the updates of $\bar{y}^{r,t}$ are given by:
$$\begin{cases} \bar{y}_1^{r,t} = y_1^{r,t} + \frac{1}{\lambda}\left(\frac{b_1}{\sigma} - A_1 R_y\right), \\ \bar{y}_2^{r,t} = \Pi_{\mathbb{R}_+^{m_2}}\left(y_2^{r,t} + \frac{1}{\lambda}\left(\frac{b_2}{\sigma} - A_2 R_y\right)\right), \end{cases}$$

where $R_y := \bar{x}^{r,t}/\sigma + (A^*y^{r,t} + \bar{z}^{r,t} - c)$. Thus, the penalty parameter $\sigma_{r+1}$ is updated as:
$$\sigma_{r+1} = \frac{1}{\sqrt{\lambda}}\frac{\|\bar{x}_a^{r,\tau_r} - x^{r,0}\|}{\|\bar{y}_a^{r,\tau_r} - y^{r,0}\|}.$$

## 4 NUMERICAL EXPERIMENT

In this section, we evaluate the performance of EPR-LP, EDR-LP, and cuPDLP (Lu & Yang, 2023) on Mittelmann's LP benchmark set[3] and LP instances relaxed from MIP problems in the MIPLIB 2017 collection (Gleixner et al., 2021). All algorithms are implemented in Julia, with experiments conducted on an NVIDIA A100-SXM4-80GB GPU running CUDA 12.3. Each solver is terminated when the relative primal and dual infeasibility errors, as well as the relative duality gap, reach a tolerance of $10^{-8}$, or when the time limit is exceeded. For more details on the experimental setup, please refer to Appendix F.

### 4.1 MITTELMANN'S LP BENCHMARK SET

Mittelmann's LP benchmark set is a standard benchmark for evaluating the numerical performance of LP solvers. In this experiment, we compare the performance of EPR-LP, EDR-LP, and cuPDLP on 49 publicly available instances from Mittelmann's LP benchmark. The performance profiles (Dolan & Moré, 2002) for solving time on the presolved and unpresolved datasets are shown in Figures 2a and 2b, respectively. The key observations are summarized as follows:

- Compared to EDR-LP and cuPDLP, EPR-LP is the fastest solver on approximately 80% of the problems in the presolved dataset and 65% in the unpresolved dataset.
- Among all the algorithms, EPR-LP demonstrates the best success rate, solving the highest percentage of problems across both the presolved and unpresolved datasets. In particular, EPR-LP solves about 8% more problems than cuPDLP on the presolved dataset.
- To solve 90% of the problems in the presolved dataset, EPR-LP requires twice the time of the best solver, while EDR-LP takes four times as long. This difference likely stems from the ratio of $\rho$, as suggested by the iteration complexity results in (13) and (14).

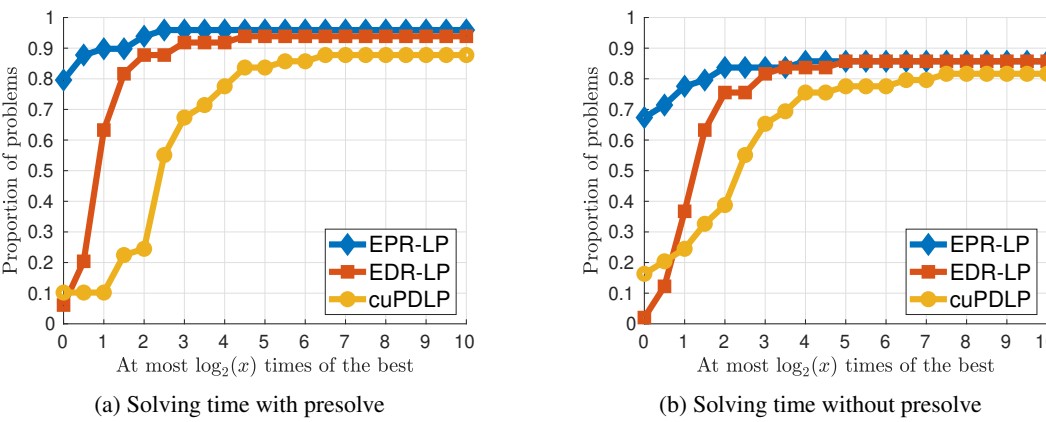

(a) Solving time with presolve

(b) Solving time without presolve

Figure 2: Performance profiles of solving time for 49 instances of Mittelmann's LP benchmark set with Gurobi's presolve (a) and without presolve (b).

### 4.2 MIP RELAXATIONS

In this experiment, we evaluate the performance of EPR-LP, EDR-LP, and cuPDLP on 380 LP instances relaxed from MIPLIB 2017 collection (Gleixner et al., 2021), both with and without presolve. Figures 3a and 3b show the performance profiles for solving time on the presolved and unpresolved datasets, respectively. The key observations are listed below:

- Compared to EDR-LP and cuPDLP, EPR-LP is the fastest solver on approximately 85% of the problems in the presolved dataset and 80% in the unpresolved dataset.
- EPR-LP demonstrated a slightly higher success rate than cuPDLP on the presolved dataset and a nearly comparable success rate on the unpresolved dataset.

---

[3]https://plato.asu.edu/ftp/lpfeas.html.

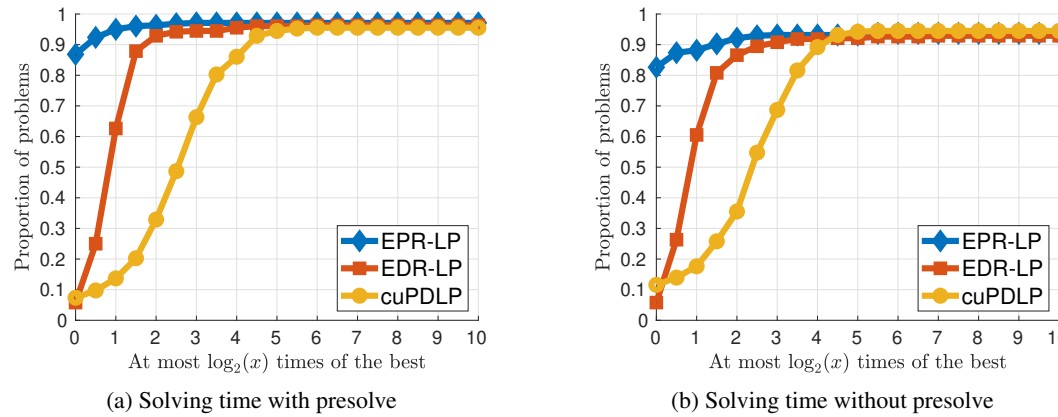

(a) Solving time with presolve        (b) Solving time without presolve

Figure 3: Performance profiles of solving time for LP instances relaxed from MIP, with Gurobi's presolve (a) and without presolve (b).

### 4.3 Summary of Experiments

The previous numerical experiments show that EPR-LP outperforms both EDR-LP and cuPDLP. Specifically, EPR-LP achieves the highest success rates across nearly all datasets. Moreover, EPR-LP emerges as the fastest solver for approximately 65% to 85% of the problems. Since the per-iteration times of EDR-LP and EPR-LP are nearly the same, EPR-LP's advantage over EDR-LP stems from using $\rho = 2$, leading to fewer iterations. To further underscore EPR-LP's advantages over cuPDLP, we present the per-iteration time ratio between these two solvers in Table 1. The median per-iteration time ratio, ranging from 2.7 to 4.0 across datasets, highlights EPR-LP's lower iteration cost compared to cuPDLP, likely attributed to cuPDLP's dependence on a more time-intensive heuristic line search.

Table 1: Per-iteration time ratio (cuPDLP/EPR-LP) for different datasets with and without presolve.

| Dataset | Median | Mean | Standard deviation |
|---|---|---|---|
| Mittelmann's LP benchmark set without presolve | 2.7 | 3.5 | 3.0 |
| Mittelmann's LP benchmark set with presolve | 3.2 | 3.9 | 4.0 |
| MIP relaxations without presolve | 3.9 | 8.2 | 14.5 |
| MIP relaxations with presolve | 4.0 | 8.8 | 15.5 |

## 5 Conclusion

In this paper, we proved the ergodic convergence of the PR method with semi-proximal terms for solving convex optimization problems. We established the ergodic iteration complexity of $O(1/k)$ with respect to the objective error, the feasibility violation, and the KKT residual based on $\varepsilon$-subdifferential. Building on these results, we developed the solver EPR-LP for solving large-scale LP problems, which incorporates adaptive restart and penalty parameter updates. Extensive numerical experiments on LP benchmark datasets highlighted the advantages of EPR-LP compared to PDLP. In the future, it would be interesting to explore combining the ergodic sequence with acceleration techniques, such as Halpern's iteration (Halpern, 1967; Lieder, 2021; Sun et al., 2024), to design a more efficient algorithm with an iteration complexity better than $O(1/k)$.

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

# A    PROOFS FOR SECTION 2

## A.1    PROOF OF PROPOSITION 2.2

*Proof.* Since $\widehat{\mathcal{T}}$ is $\mathcal{M}$-firmly nonexpansive, as stated in Sun et al. (2024, Proposition 2.3), we obtain

$$\left\| w^{k+1} - w^* \right\|_{\mathcal{M}}^2 \le \left\| w^k - w^* \right\|_{\mathcal{M}}^2 - \rho(2 - \rho) \left\| w^k - \bar{w}^k \right\|_{\mathcal{M}}^2, \quad \forall k \ge 0, w^* \in \mathcal{T}^{-1}(0), \quad (22)$$

which, together with the iteration scheme (8), implies that

$$
\begin{aligned}
\|\bar{w}_a^k - w_a^k\|_{\mathcal{M}} &= \left\| \frac{1}{k+1} \sum_{t=0}^{k} (\bar{w}^t - w^t) \right\|_{\mathcal{M}} \\
&= \left\| \frac{1}{k+1} \sum_{t=0}^{k} \frac{(w^{t+1} - w^t)}{\rho} \right\|_{\mathcal{M}} \\
&= \frac{1}{\rho(k+1)} \left\| (w^{k+1} - w^0) \right\|_{\mathcal{M}} \\
&\le \frac{1}{\rho(k+1)} \left\| (w^{k+1} - w^*) - (w^0 - w^*) \right\|_{\mathcal{M}} \\
&\le \frac{2}{\rho(k+1)} \left\| (w^0 - w^*) \right\|_{\mathcal{M}}, \quad \forall k \ge 0, w^* \in \mathcal{T}^{-1}(0).
\end{aligned}
$$

This concludes the proof of statement (a). Moreover, for any $(w', v') \in \mathrm{gph}(\mathcal{T})$, and by the definitions of $w_a^k$ and $\bar{w}_a^k$ in (9), we have, for any $k \geq 0$,

$$
\begin{aligned}
&\langle \bar{w}_a^k - w', \mathcal{M}(w_a^k - \bar{w}_a^k) - v' \rangle \\
=\ & \tfrac{1}{k+1} \sum_{t=0}^{k} \langle \bar{w}^t - w', \mathcal{M}(w_a^k - \bar{w}_a^k) - v' \rangle \\
=\ & \tfrac{1}{k+1} \sum_{t=0}^{k} \left( \langle \bar{w}^t - w', \mathcal{M}(w_a^k - \bar{w}_a^k) - \mathcal{M}(w^t - \bar{w}^t) \rangle + \langle \bar{w}^t - w', \mathcal{M}(w^t - \bar{w}^t) - v' \rangle \right) \\
\geq\ & \tfrac{1}{k+1} \sum_{t=0}^{k} \left( \langle \bar{w}^t - w', \mathcal{M}(w_a^k - \bar{w}_a^k) - \mathcal{M}(w^t - \bar{w}^t) \rangle \right) \\
=\ & \tfrac{1}{k+1} \sum_{t=0}^{k} \left( \langle \bar{w}^t - \bar{w}_a^k, \mathcal{M}(w_a^k - \bar{w}_a^k) - \mathcal{M}(w^t - \bar{w}^t) \rangle + \langle \bar{w}_a^k - w', \mathcal{M}(w_a^k - \bar{w}_a^k) - \mathcal{M}(w^t - \bar{w}^t) \rangle \right) \\
=\ & \tfrac{1}{k+1} \sum_{t=0}^{k} \left( \langle \bar{w}^t - \bar{w}_a^k, (w_a^k - \bar{w}_a^k) - (w^t - \bar{w}^t) \rangle_{\mathcal{M}} \right) \\
=\ & -\tfrac{1}{k+1} \sum_{t=0}^{k} \left( \langle \bar{w}^t - \bar{w}_a^k, (w^t - \bar{w}^t) \rangle_{\mathcal{M}} \right) \\
=\ & -\bar{\varepsilon}_a^k.
\end{aligned}
$$

Next, we prove that $\bar{\varepsilon}_a^k \geq 0$ for all $k \geq 0$ by contradiction. Suppose $\bar{\varepsilon}_a^k < 0$ for some $k \geq 0$. Then, for any $(w', v') \in \mathrm{gph}(\mathcal{T})$, we have

$$
\langle \bar{w}_a^k - w', \mathcal{M}(w_a^k - \bar{w}_a^k) - v' \rangle > 0,
$$

which, combined with the maximality of $\mathcal{T}$, implies that $(\bar{w}_a^k, \mathcal{M}(w_a^k - \bar{w}_a^k)) \in \mathrm{gph}(\mathcal{T})$. Taking $(w', v') = (\bar{w}_a^k, \mathcal{M}(w_a^k - \bar{w}_a^k))$, we obtain $0 \geq -\bar{\varepsilon}_a^k$, which contradicts our assumption. Thus, $\bar{\varepsilon}_a^k \geq 0$ for all $k \geq 0$. Next, we establish an upper bound for $\bar{\varepsilon}_a^k$ for any $k \geq 0$. Indeed, we have

$$
\begin{aligned}
\bar{\varepsilon}_a^k &= \tfrac{1}{k+1} \sum_{t=0}^{k} \langle w^t - \bar{w}^t, \bar{w}^t - \bar{w}_a^k \rangle_{\mathcal{M}} \\
&= \tfrac{1}{k+1} \sum_{t=0}^{k} \left( \langle \tfrac{1}{\rho}(w^t - w^{t+1}), w^t - \tfrac{1}{\rho}(w^t - w^{t+1}) - \bar{w}_a^k \rangle_{\mathcal{M}} \right) \\
&= \tfrac{1}{k+1} \sum_{t=0}^{k} \left( -\tfrac{1}{\rho^2} \|w^t - w^{t+1}\|_{\mathcal{M}}^2 + \tfrac{1}{2\rho}(\|w^t - w^{t+1}\|_{\mathcal{M}}^2 + \|w^t - \bar{w}_a^k\|_{\mathcal{M}}^2 - \|w^{t+1} - \bar{w}_a^k\|_{\mathcal{M}}^2) \right) \\
&\leq \tfrac{1}{k+1} \sum_{t=0}^{k} \left( \tfrac{1}{2\rho}(\|w^t - \bar{w}_a^k\|_{\mathcal{M}}^2 - \|w^{t+1} - \bar{w}_a^k\|_{\mathcal{M}}^2) \right) \\
&= \tfrac{1}{2\rho(k+1)} \left( \|w^0 - \bar{w}_a^k\|_{\mathcal{M}}^2 - \|w^{k+1} - \bar{w}_a^k\|_{\mathcal{M}}^2 \right) \\
&= \tfrac{1}{2\rho(k+1)} \left( -\|w^0 - w^{k+1}\|_{\mathcal{M}}^2 - 2\langle w^{k+1} - w^0, w^0 - \bar{w}_a^k \rangle_{\mathcal{M}} \right) \\
&\leq \tfrac{1}{2\rho(k+1)} \left( -\|w^0 - w^{k+1}\|_{\mathcal{M}}^2 + 2\|w^{k+1} - w^0\|_{\mathcal{M}} \|w^0 - \bar{w}_a^k\|_{\mathcal{M}} \right).
\end{aligned}
\tag{23}
$$

Using the convexity of $\|\cdot\|_{\mathcal{M}}$, we can obtain

$$
\|w^0 - w_a^k\|_{\mathcal{M}} = \|\tfrac{1}{k+1} \sum_{t=0}^{k} (w^0 - w^t)\|_{\mathcal{M}} \leq \tfrac{1}{k+1} \sum_{t=0}^{k} \|w^0 - w^t\|_{\mathcal{M}} \leq 2\|w^0 - w^*\|_{\mathcal{M}}.
$$

Combining this with (23), we derive that for any $k \geq 0$,

$$
\begin{aligned}
\bar{\varepsilon}_a^k &\leq \tfrac{1}{2\rho(k+1)} \left( -\|w^0 - w^{k+1}\|_{\mathcal{M}}^2 + 2\|w^{k+1} - w^0\|_{\mathcal{M}} \|w^0 - w^*\|_{\mathcal{M}} \right) \\
&\leq \tfrac{1}{2\rho(k+1)} \|w^0 - w^*\|_{\mathcal{M}}^2.
\end{aligned}
$$

This completes the proof. $\qquad \square$

### A.2 Proof of Theorem 2.1

*Proof.* Suppose that $\mathcal{M} = \mathcal{C}\mathcal{C}^*$ is a decomposition of $\mathcal{M}$ according to Bredies et al. (2022, Proposition 2.3), where $\mathcal{C} : \mathbb{U} \to \mathbb{W}$. Since $(\mathcal{M} + \mathcal{T})^{-1}$ is $L$-Lipschitz continuous and $\|\mathcal{C}^* w\| = \|w\|_{\mathcal{M}}$ for every $w \in \mathcal{H}$, we have, for all $w' \in \mathcal{H}$ and $w^* \in \mathcal{T}^{-1}(0)$,

$$
\|\widehat{\mathcal{T}} w' - \widehat{\mathcal{T}} w^*\| = \|(\mathcal{M} + \mathcal{T})^{-1} \mathcal{C}\mathcal{C}^* w' - (\mathcal{M} + \mathcal{T})^{-1} \mathcal{C}\mathcal{C}^* w^*\| \leq L \|\mathcal{C}\| \|w' - w^*\|_{\mathcal{M}},
$$

which, together with (22), implies that

$$
\|\bar{w}^k - w^*\| = \|\widehat{\mathcal{T}} w^k - w^*\| \leq L \|\mathcal{C}\| \|w^k - w^*\|_{\mathcal{M}} \leq L \|\mathcal{C}\| \|w^0 - w^*\|_{\mathcal{M}}.
$$

Thus, both sequences $\{\bar{w}^k\}$ and $\{\bar{w}_a^k\}$ are bounded. According to Proposition 2.2 and the maximality of $\mathcal{T}$, any cluster point of $\{\bar{w}_a^k\}$ belongs to $\mathcal{T}^{-1}(0)$.

To establish the uniqueness of cluster points, we define two shadow sequences as follows:

$$
u^k := \mathcal{C}^* w^k \quad \text{and} \quad u_a^k := \frac{1}{k+1} \sum_{t=0}^{k} u^t, \quad \forall k \geq 0. \tag{24}
$$

Through straightforward calculations, we obtain

$$u^{k+1} = \widehat{\mathcal{F}}_\rho u^k, \quad \forall k \geq 0,$$

where $\widehat{\mathcal{F}}_\rho := (1-\rho)\mathcal{I} + \rho(\mathcal{C}^*(\mathcal{M}+\mathcal{T})^{-1}\mathcal{C})$ with $\rho \in (0,2]$ is a nonexpansive operator, according to Sun et al. (2024, Proposition 2.5). By Baillon's nonlinear ergodic theorem (Baillon, 1975) and (Bauschke & Combettes, 2017, Example 5.38), the sequence $\{u_a^k\}$ converges to a point in $\text{Fix}(\widehat{\mathcal{F}}_\rho)$, where $\text{Fix}(\widehat{\mathcal{F}}_\rho)$ represents the set of fixed points of the nonexpansive operator $\widehat{\mathcal{F}}_\rho$. Given the equivalence between $\text{Fix}(\widehat{\mathcal{F}}_\rho)$ and $\mathcal{C}^*\mathcal{T}^{-1}(0)$ as stated in Sun et al. (2024, Proposition 2.5), we conclude that there exists $w_a^* \in \mathcal{T}^{-1}(0)$ such that

$$\|u_a^k - \mathcal{C}^* w_a^*\| \to 0.$$

Therefore, by the definition of $\{u_a^k\}$ in (24), we have

$$\|w_a^k - w_a^*\|_\mathcal{M} = \left\| \frac{1}{k+1} \sum_{t=0}^k \mathcal{C}^* w^t - \mathcal{C}^* w_a^* \right\| = \|u_a^k - \mathcal{C}^* w_a^*\| \to 0,$$

which, together with part (a) of Proposition 2.2, implies that

$$\|\bar{w}_a^k - w_a^*\|_\mathcal{M}^2 = \|\bar{w}_a^k - w_a^k\|_\mathcal{M}^2 + \|w_a^k - w_a^*\|_\mathcal{M}^2 + 2\langle \bar{w}_a^k - w_a^k, w^k - w_a^* \rangle_\mathcal{M} \to 0. \quad (25)$$

Since the sequence $\{\bar{w}_a^k\}$ is bounded, it must have at least one cluster point. Assume a subsequence $\{\bar{w}_a^{k_i}\}$ converges to $w^*$. Suppose $\|w^* - w_a^*\|_\mathcal{M} > 0$. By an Opial-type argument (Opial, 1967, Lemma 1), we have

$$\liminf_{i\to\infty} \|\bar{w}_a^{k_i} - w^*\|_\mathcal{M} < \liminf_{i\to\infty} \|\bar{w}_a^{k_i} - w_a^*\|_\mathcal{M},$$

which, combined with (25), implies $\liminf_{i\to\infty} \|\bar{w}_a^{k_i} - w^*\|_\mathcal{M} < 0$, a contradiction to the semi-positive definiteness of $\mathcal{M}$. Hence, $\|w^* - w_a^*\|_\mathcal{M} = 0$. It follows that

$$w^* = (\mathcal{M}+\mathcal{T})^{-1}\mathcal{M}w^* = (\mathcal{M}+\mathcal{T})^{-1}\mathcal{M}w_a^* = w_a^*.$$

Taking any other cluster point $w^{**}$, we can similarly show that $w^{**} = w_a^*$. Therefore, the cluster point is unique, and the sequence $\{\bar{w}_a^k\}$ converges to $w_a^*$. $\qquad\square$

### A.3 PROOF OF LEMMA 2.1

*Proof.* From the optimality conditions of the subproblems in Algorithm 2, we have, for any $t \geq 0$,

$$\begin{cases} f_2(z) \geq f_2(\bar{z}^t) + \langle -B_2^* \bar{x}^t - \mathcal{T}_2(\bar{z}^t - z^t), z - \bar{z}^t \rangle, & \forall z \in \mathbb{Z}, \\ f_1(y) \geq f_1(\bar{y}^t) + \langle -B_1^*(\bar{x}^t + \sigma(B_1\bar{y}^t + B_2\bar{z}^t - c)) - \mathcal{T}_1(\bar{y}^t - y^t), y - \bar{y}^t \rangle, & \forall y \in \mathbb{Y}. \end{cases}$$

Summing these from $t = 0$ to $k$, and dividing by $k+1$, we obtain

$$\begin{cases} f_2(z) \geq \frac{1}{k+1}\sum_{t=0}^k f_2(\bar{z}^t) + \frac{1}{k+1}\sum_{t=0}^k \langle -B_2^* \bar{x}^t - \mathcal{T}_2(\bar{z}^t - z^t), z - \bar{z}^t \rangle, & \forall z \in \mathbb{Z}, \\ f_1(y) \geq \frac{1}{k+1}\sum_{t=0}^k f_1(\bar{y}^t) + \frac{1}{k+1}\sum_{t=0}^k \langle -B_1^*(\bar{x}^t + \sigma(B_1\bar{y}^t + B_2\bar{z}^t - c)) \\ \qquad\qquad -\mathcal{T}_1(\bar{y}^t - y^t), y - \bar{y}^t \rangle, & \forall y \in \mathbb{Y}. \end{cases}$$

By the convexity of $f_2$, we have

$$\begin{aligned} f_2(z) &\geq f_2(\bar{z}_a^k) + \frac{1}{k+1}\sum_{t=0}^k \langle -B_2^* \bar{x}^t - \mathcal{T}_2(\bar{z}^t - z^t), z - \bar{z}^t \rangle, \forall z \in \mathbb{Z} \\ &= f_2(\bar{z}_a^k) + \frac{1}{k+1}\sum_{t=0}^k \langle -B_2^* \bar{x}^t - \mathcal{T}_2(\bar{z}^t - z^t), z - \bar{z}_a^k \rangle - \bar{\varepsilon}_z^k, \forall z \in \mathbb{Z}, \end{aligned}$$

where

$$\bar{\varepsilon}_z^k := \frac{1}{k+1}\sum_{t=0}^k \langle -B_2^* \bar{x}^t - \mathcal{T}_2(\bar{z}^t - z^t), \bar{z}^t - \bar{z}_a^k \rangle$$

is non-negative by substituting $z = \bar{z}_a^k$ in the first inequality. Hence, we have

$$-(B_2^* \bar{x}_a^k + \mathcal{T}_2(\bar{z}_a^k - z_a^k)) = \frac{1}{k+1}\sum_{t=0}^k -(B_2^* \bar{x}^t + \mathcal{T}_2(\bar{z}^t - z^t)) \in \partial_{\bar{\varepsilon}_z^k} f_2(\bar{z}_a^k), \forall k \geq 0.$$

Similarly, we obtain

$$-B_1^*(\bar{x}_a^k + \sigma(B_1\bar{y}_a^k + B_2\bar{z}_a^k - c)) - \mathcal{T}_1(\bar{y}_a^k - y_a^k) \in \partial_{\bar{\varepsilon}_y^k} f_1(\bar{y}_a^k), \ \forall k \geq 0,$$

where

$$\bar{\varepsilon}_y^k := \frac{1}{k+1} \sum_{t=0}^k \langle -B_1^*(\bar{x}^t + \sigma(B_1\bar{y}^t + B_2\bar{z}^t - c)) - \mathcal{T}_1(\bar{y}^t - y^t), \bar{y}^t - \bar{y}_a^k \rangle \geq 0.$$

Now, we show the upper bound of $\bar{\varepsilon}_z^k + \bar{\varepsilon}_y^k$. According to definitions of $\bar{\varepsilon}_z^k$ and $\bar{\varepsilon}_y^k$ in (11), we have

$$
\begin{aligned}
&\bar{\varepsilon}_z^k + \bar{\varepsilon}_y^k \\
&= \tfrac{1}{k+1} \sum_{t=0}^k \left( \langle -B_2^*\bar{x}^t - \mathcal{T}_2(\bar{z}^t - z^t), \bar{z}^t - \bar{z}_a^k \rangle + \langle -B_1^*(\bar{x}^t + \sigma(B_1\bar{y}^t + B_2\bar{z}^t - c)) - \mathcal{T}_1(\bar{y}^t - y^t), \bar{y}^t - \bar{y}_a^k \rangle \right) \\
&= \tfrac{1}{k+1} \sum_{t=0}^k \left( \langle \mathcal{M}(w^t - \bar{w}^t), \bar{w}^t - \bar{w}_a^k \rangle - \langle B_1^*\bar{x}^t, y^t - \bar{y}_a^k \rangle - \langle B_2^*\bar{x}^t, z^t - \bar{z}_a^k \rangle - \langle c - B_1\bar{y}^t - B_2\bar{z}^t, \bar{x}^t - \bar{x}_a^k \rangle \right) \\
&= \tfrac{1}{k+1} \sum_{t=0}^k \left( \langle \mathcal{M}(w^t - \bar{w}^t), \bar{w}^t - \bar{w}_a^k \rangle \right).
\end{aligned}
$$

Thus, by the definition of $\bar{\varepsilon}_a^k$ in Proposition 2.2, and using the equivalence between the pADMM and the dPPM in Proposition 2.1, we can derive

$$\bar{\varepsilon}_z^k + \bar{\varepsilon}_y^k = \bar{\varepsilon}_a^k \leq \frac{1}{2\rho(k+1)} \|w^0 - w^*\|_{\mathcal{M}}^2.$$

This completes the proof. $\qquad\square$

### A.4 PROOF OF THEOREM 2.2

*Proof.* According to Propositions 2.1 and 2.2, we have

$$\|\bar{w}_a^k - w_a^k\|_{\mathcal{M}}^2 \leq \frac{4R_0^2}{\rho^2(k+1)^2}, \quad \forall k \geq 0.$$

By the definition of $\mathcal{M}$ in (7), this can be rewritten as

$$\|\bar{y}_a^k - y_a^k\|_{\mathcal{T}_1}^2 + \frac{1}{\sigma} \|\sigma B_1(\bar{y}_a^k - y_a^k) + (\bar{x}_a^k - x_a^k)\|^2 + \|\bar{z}_a^k - z_a^k\|_{\mathcal{T}_2}^2 \leq \frac{4R_0^2}{\rho^2(k+1)^2}, \ \forall k \geq 0. \quad (26)$$

From Step 2 of Algorithm 2, we can deduce that for any $k \geq 0$,

$$
\begin{aligned}
\|\sigma B_1(\bar{y}_a^k - y_a^k) + (\bar{x}_a^k - x_a^k)\| &= \|\sigma B_1(\bar{y}_a^k - y_a^k) + \sigma(B_1 y_a^k + B_2 \bar{z}_a^k - c)\| \\
&= \sigma \|B_1\bar{y}_a^k + B_2\bar{z}_a^k - c\|,
\end{aligned}
$$

which, together with (26), yields that

$$\|B_1\bar{y}_a^k + B_2\bar{z}_a^k - c\| \leq \frac{2R_0}{\sqrt{\sigma}\rho(k+1)}, \quad \forall k \geq 0. \quad (27)$$

Furthermore, according to the Lemma 2.1, we have for $k \geq 0$,

$$
\begin{cases}
-B_2^*\bar{x}_a^k - \mathcal{T}_2(\bar{z}_a^k - z_a^k) \in \partial_{\bar{\varepsilon}_z^k} f_2(\bar{z}_a^k), \\
-B_1^*(\bar{x}_a^k + \sigma(B_1\bar{y}_a^k + B_2\bar{z}_a^k - c)) - \mathcal{T}_1(\bar{y}_a^k - y_a^k) \in \partial_{\bar{\varepsilon}_y^k} f_1(\bar{y}_a^k),
\end{cases}
$$

which, together with (26) and (27), implies

$$\text{dist}\left(0, \partial_{\bar{\varepsilon}_z^k} f_2(\bar{z}_a^k) + B_2^*\bar{x}_a^k\right) \leq \|\mathcal{T}_2(\bar{z}_a^k - z_a^k)\| \leq \|\sqrt{\mathcal{T}_2}\| \|(\bar{z}_a^k - z_a^k)\|_{\mathcal{T}_2} \leq \|\sqrt{\mathcal{T}_2}\| \frac{2R_0}{\rho(k+1)} \quad (28)$$

and

$$
\begin{aligned}
\text{dist}\left(0, \partial_{\bar{\varepsilon}_y^k} f_1(\bar{y}_a^k) + B_1^*\bar{x}_a^k\right) &\leq \sigma\|B_1^*\| \|(B_1\bar{y}_a^k + B_2\bar{z}_a^k - c)\| + \|\mathcal{T}_1(\bar{y}_a^k - y_a^k)\| \\
&\leq \left(\frac{\sigma\|B_1^*\|}{\sqrt{\sigma}} + \|\sqrt{\mathcal{T}_1}\|\right) \frac{2R_0}{\rho(k+1)}.
\end{aligned} \quad (29)
$$

Thus, combining (27), (28), (29) and Lemma 2.1, we derive the iteration complexity bound in (13).

We now estimate the ergodic iteration complexity results for the objective error. From the KKT conditions in (5), we have, for any $k \geq 0$,

$$f_1(\bar{y}_a^k) - f_1(y^*) \geq \langle -B_1^* x^*, \bar{y}_a^k - y^* \rangle, \quad f_2(\bar{z}_a^k) - f_2(z^*) \geq \langle -B_2^* x^*, \bar{z}_a^k - z^* \rangle.$$

Thus, it follows from (27) that for all $k \geq 0$,

$$\begin{aligned}
h(\bar{y}_a^k, \bar{z}_a^k) &\geq \langle B_1 \bar{y}_a^k + B_2 \bar{z}_a^k - c, -x^* \rangle \\
&\geq -\|x^*\| \|B_1 \bar{y}_a^k + B_2 \bar{z}_a^k - c\| \\
&\geq -\frac{2R_0 \|x^*\|}{\sqrt{\sigma} \rho(k+1)}.
\end{aligned}$$

For the upper bound of the objective error, from Sun et al. (2024, Lemma 3.6), we first have the following upper bounds:

$$\begin{aligned}
h(\bar{y}^k, \bar{z}^k) &\leq \langle \sigma B_1(y^* - \bar{y}^k) - \bar{x}^k, (B_1 \bar{y}^k + B_2 \bar{z}^k - c) \rangle \\
&\quad + \langle y^* - \bar{y}^k, \mathcal{T}_1(\bar{y}^k - y^k) \rangle + \langle z^* - \bar{z}^k, \mathcal{T}_2(\bar{z}^k - z^k) \rangle.
\end{aligned} \tag{30}$$

Note that from Step 4 of Algorithm 2, we have for any $k \geq 0$,

$$\begin{aligned}
&\langle y^* - \bar{y}^k, \mathcal{T}_1(\bar{y}^k - y^k) \rangle \\
&= \langle y^* - (y^k + \frac{y^{k+1} - y^k}{\rho}), \mathcal{T}_1(\frac{y^{k+1} - y^k}{\rho}) \rangle \\
&= \frac{1}{2\rho}(\|y^k - y^*\|_{\mathcal{T}_1}^2 - \|y^{k+1} - y^*\|_{\mathcal{T}_1}^2) + \frac{\rho - 2}{2\rho^2}\|y^{k+1} - y^k\|_{\mathcal{T}_1}^2 \\
&\leq \frac{1}{2\rho}(\|y^k - y^*\|_{\mathcal{T}_1}^2 - \|y^{k+1} - y^*\|_{\mathcal{T}_1}^2).
\end{aligned} \tag{31}$$

Similarly, we also have

$$\langle z^* - \bar{z}^k, \mathcal{T}_2(\bar{z}^k - z^k) \rangle \leq \frac{1}{2\rho}(\|z^k - z^*\|_{\mathcal{T}_1}^2 - \|z^{k+1} - z^*\|_{\mathcal{T}_1}^2), \quad \forall k \geq 0. \tag{32}$$

Additionally, define

$$\Delta_k := x^k + \sigma B_1 y^k, \quad \forall k \geq 0.$$

From Step 4 of Algorithm 2, we can derive that

$$\begin{aligned}
&\langle \sigma B_1(y^* - \bar{y}^k) - \bar{x}^k, (B_1 \bar{y}^k + B_2 \bar{z}^k - c) \rangle \\
&= \langle \sigma B_1 y^*, (B_1 \bar{y}^k + B_2 \bar{z}^k - c) \rangle - \langle \bar{x}^k + \sigma B_1 \bar{y}^k, (B_1 \bar{y}^k + B_2 \bar{z}^k - c) \rangle \\
&= \langle \sigma B_1 y^*, (B_1 \bar{y}^k + B_2 \bar{z}^k - c) \rangle - \langle \Delta_k + \frac{\Delta_{k+1} - \Delta_k}{\rho}, \frac{\Delta_{k+1} - \Delta_k}{\rho} \rangle \\
&= \langle \sigma B_1 y^*, (B_1 \bar{y}^k + B_2 \bar{z}^k - c) \rangle - \frac{1}{2\rho}(\|\Delta_{k+1}\|^2 - \|\Delta_k\|^2) + \frac{\rho - 2}{2\rho^2}\|\Delta_{k+1}\|^2 \\
&\leq \langle \sigma B_1 y^*, (B_1 \bar{y}^k + B_2 \bar{z}^k - c) \rangle - \frac{1}{2\rho}(\|\Delta_{k+1}\|^2 - \|\Delta_k\|^2), \quad \forall k \geq 0.
\end{aligned} \tag{33}$$

Thus, combing with (30), (31), (32), and (33), we conclude that for all $k \geq 0$,

$$\begin{aligned}
h(\bar{y}^k, \bar{z}^k) &\leq \frac{1}{2\rho}(\|y^k - y^*\|_{\mathcal{T}_1}^2 - \|y^{k+1} - y^*\|_{\mathcal{T}_1}^2) + \frac{1}{2\rho}(\|z^k - z^*\|_{\mathcal{T}_1}^2 - \|z^{k+1} - z^*\|_{\mathcal{T}_1}^2) \\
&\quad + \langle \sigma B_1 y^*, (B_1 \bar{y}^k + B_2 \bar{z}^k - c) \rangle - \frac{1}{2\rho}(\|\Delta_{k+1}\|^2 - \|\Delta_k\|^2).
\end{aligned}$$

It follows from the convexity of $h$ and (27) that for any $k \geq 0$,

$$\begin{aligned}
h(\bar{y}_a^k, \bar{z}_a^k) &\leq \frac{1}{k+1} \sum_{t=0}^{k} h(\bar{y}^t, \bar{z}^t) \\
&\leq \frac{1}{2\rho(k+1)}\left(\|y^0 - y^*\|_{\mathcal{T}_1}^2 + \|z^0 - z^*\|_{\mathcal{T}_2}^2\right) + \langle \sigma B_1 y^*, (B_1 \bar{y}_a^k + B_2 \bar{z}_a^k - c) \rangle + \frac{1}{2\rho(k+1)}\|\Delta_0\|^2 \\
&\leq \frac{R_0^2}{2\rho(k+1)} + \sqrt{\sigma}\|B_1 y^*\| \frac{2R_0}{\rho(k+1)} + \frac{1}{2\rho(k+1)}\|x^0 + \sigma B_1 y^0\|^2.
\end{aligned}$$

This completes the proof. $\qquad \square$

## B   THE ERGODIC ITERATION COMPLEXITY OF PADMM FOR SOLVING LP

In this section, without loss of generality, we consider the LP problem without inequality constraints (i.e., $m_2 = 0$). Inequality constraints can be easily converted into equality constraints by introducing slack variables. The KKT system of the LP problem (2) is

$$0 \in Ax^* - b, \quad 0 \in z^* + \partial \delta_C(x^*), \quad A^* y^* + z^* - c = 0. \tag{34}$$

Let $f_1(y) = \langle b, y \rangle$, $f_2(z) = \delta_C^*(-z)$, $B_1 = A^*, B_2 = I_n$, $\mathcal{T}_1 = \sigma(\lambda_1(AA^*)I_m - AA^*)$, $\mathcal{T}_2 = 0$. Then, for LP we define

$$
\mathcal{M}^{LP} = \begin{bmatrix} \sigma\lambda_1(AA^*)I_m & 0 & A^* \\ 0 & 0 & 0 \\ A & 0 & \sigma^{-1}I_n \end{bmatrix},
$$

and

$$
h^{LP}(\bar{y}_a^k, \bar{z}_a^k) := \langle b, y_a^k \rangle + \delta_C^*(-z_a^k) - (\langle b, y^* \rangle + \delta_C^*(-z^*)), \quad \forall k \geq 0,
$$

where $(y^*, z^*)$ is a solution to the problem (2). Thus, the ergodic complexity of pADMM for solving the LP problem (2) is presented in the following corollary.

**Corollary B.1.** *Suppose the solution set to the KKT system (34) is nonempty. Let $\{(\bar{y}_a^k, \bar{z}_a^k, \bar{x}_a^k)\}$ be the ergodic sequence generated by Algorithm 2 with $\mathcal{T}_1 = \sigma(\lambda_1(AA^*)I_m - AA^*)$, $\mathcal{T}_2 = 0$, and $\rho \in (0, 2]$. Let $w^* = (y^*, z^*, x^*)$ be a solution to the KKT system (34), and define $R_0 = \|w^0 - w^*\|_{\mathcal{M}^{LP}}$. For all $k \geq 0$, the following ergodic iteration complexity bound holds:*

$$
\begin{cases} \|A\bar{x}_a^k - b\| + \|A^*\bar{y}_a^k + \bar{z}_a^k - c\| \leq \left( \dfrac{2\sigma\sqrt{\lambda_1(AA^*)} + 1}{\sqrt{\sigma}} \right) \dfrac{2R_0}{\rho(k+1)}, \\ -\bar{z}_a^k \in \partial_{\bar{\varepsilon}_z^k}\delta_C(\bar{x}_a^k), \end{cases}
$$

*where $\partial_{\bar{\varepsilon}_z^k}\delta_C(\bar{x}_a^k) = \{z \in \mathbb{R}^n \mid \langle z, x' - \bar{x}_a^k \rangle \leq \bar{\varepsilon}_z^k, \forall x' \in C\}$ and $\bar{\varepsilon}_z^k \leq \frac{1}{2\rho(k+1)}R_0^2$. Moreover,*

$$
\left( -\frac{1}{\sqrt{\sigma}}\|x^*\| \right) \frac{2R_0}{\rho(k+1)} \leq h(\bar{y}_a^k, \bar{z}_a^k) \leq \left( R_0 + 4\sqrt{\sigma}\|A^*y^*\| \right) \frac{R_0}{2\rho(k+1)} + \frac{\|x^0 + \sigma A^*y^0\|^2}{2\rho(k+1)}.
$$

*Proof.* Furthermore, since $f_1(y) = \langle b, y \rangle$ for any $y \in \mathbb{R}^m$, it follows from (Hiriart-Urruty & Lemaréchal, 1993, Proposition 1.3.1) that, for any $\varepsilon \geq 0$,

$$
\partial_\varepsilon f_1(y) = b. \tag{35}
$$

On the other hand, from (10) and $\mathcal{T}_2 = 0$, we have

$$
\bar{x}_a^k \in \partial_{\bar{\varepsilon}_z^k}\delta_C^*(-\bar{z}_a^k),
$$

which, by (Hiriart-Urruty & Lemaréchal, 1993, Proposition 1.2.1), is equivalent to

$$
-\bar{z}_a^k \in \partial_{\bar{\varepsilon}_z^k}\delta_C(\bar{x}_a^k). \tag{36}
$$

Combing (35),(36), and Theorem 2.2, we complete the proof.

$\square$

## C  THE NON-ERGODIC ITERATION COMPLEXITY OF THE PADMM

To establish the non-ergodic iteration complexity of the pADMM, we first establish the non-ergodic iteration complexity of the dPPM.

**Proposition C.1.** *Assume that $\mathcal{T}^{-1}(0)$ is nonempty and $\rho \in (0, 2)$, then the sequences $\{w^k\}$ and $\{\bar{w}^k\}$ generated by the dPPM scheme (8) satisfy*

$$
\|w^k - \bar{w}^k\|_{\mathcal{M}}^2 \leq \frac{1}{\rho(2-\rho)(k+1)}\|w^0 - w^*\|_{\mathcal{M}}^2, \quad \forall k \geq 0, w^* \in \mathcal{T}^{-1}(0).
$$

*Furthermore, $\|w^t - \bar{w}^t\|_{\mathcal{M}}^2 = o(1/(k+1))$ as $k \to \infty$.*

*Proof.* Since $\widehat{\mathcal{T}}$ is $\mathcal{M}$-firmly nonexpansive in Sun et al. (2024, Proposition 2.3), we can obtain

$$
\|w^{k+1} - w^*\|_{\mathcal{M}}^2 \leq \|w^k - w^*\|_{\mathcal{M}}^2 - \rho(2-\rho)\|w^k - \bar{w}^k\|_{\mathcal{M}}^2, \quad \forall k \geq 0, w^* \in \mathcal{T}^{-1}(0). \tag{37}
$$

Summing this inequality from $0$ to $k$, we have

$$\sum_{t=0}^{k} \rho(2-\rho)\|w^t - \bar{w}^t\|_{\mathcal{M}}^2 \leq \|w^0 - w^*\|_{\mathcal{M}}^2 - \|w^{k+1} - w^*\|_{\mathcal{M}}^2 \leq \|w^0 - w^*\|_{\mathcal{M}}^2, \quad (38)$$

which implies that $\|w^t - \bar{w}^t\|_{\mathcal{M}}^2 = o(1/(k+1))$ as $k \to \infty$. Now, we claim that the sequence $\{\|w^t - \bar{w}^t\|_{\mathcal{M}}\}$ is monotonically nonincreasing. Indeed,

$$
\begin{aligned}
&\|w^{t+1} - \bar{w}^{t+1}\|_{\mathcal{M}}^2 \\
&= \|(w^{t+1} - \bar{w}^{t+1}) - (w^t - \bar{w}^t)\|_{\mathcal{M}}^2 + \|(w^t - \bar{w}^t)\|_{\mathcal{M}}^2 \\
&\quad + 2\langle (w^{t+1} - \bar{w}^{t+1}) - (w^t - \bar{w}^t), (w^t - \bar{w}^t)\rangle_{\mathcal{M}} \\
&= \|(w^{t+1} - \bar{w}^{t+1}) - (w^t - \bar{w}^t)\|_{\mathcal{M}}^2 + \|(w^t - \bar{w}^t)\|_{\mathcal{M}}^2 \\
&\quad - \frac{2}{\rho}\langle (w^{t+1} - \bar{w}^{t+1}) - (w^t - \bar{w}^t), (w^{t+1} - w^t)\rangle_{\mathcal{M}}.
\end{aligned}
\quad (39)
$$

Since

$$
\begin{aligned}
&\langle (w^{t+1} - \bar{w}^{t+1}) - (w^t - \bar{w}^t), (w^{t+1} - w^t)\rangle_{\mathcal{M}} \\
&= \langle (w^{t+1} - \bar{w}^{t+1}) - (w^t - \bar{w}^t), (w^{t+1} - \bar{w}^{t+1}) - (w^t - \bar{w}^t)\rangle_{\mathcal{M}} \\
&\quad + \langle (w^{t+1} - \bar{w}^{t+1}) - (w^t - \bar{w}^t), \bar{w}^{t+1} - \bar{w}^t\rangle_{\mathcal{M}} \\
&= \|(w^{t+1} - \bar{w}^{t+1}) - (w^t - \bar{w}^t)\|_{\mathcal{M}}^2 + \langle (w^{t+1} - \bar{w}^{t+1}) - (w^t - \bar{w}^t), \bar{w}^{t+1} - \bar{w}^t\rangle_{\mathcal{M}},
\end{aligned}
$$

we can obtain from (39) that

$$
\begin{aligned}
&\|w^{t+1} - \bar{w}^{t+1}\|_{\mathcal{M}}^2 \\
&= \|w^t - \bar{w}^t\|_{\mathcal{M}}^2 - \frac{2-\rho}{\rho}\|(w^{t+1} - \bar{w}^{t+1}) - (w^t - \bar{w}^t)\|_{\mathcal{M}}^2 \\
&\quad - \frac{2}{\rho}\langle (w^{t+1} - \bar{w}^{t+1}) - (w^t - \bar{w}^t), \bar{w}^{t+1} - \bar{w}^t\rangle_{\mathcal{M}}.
\end{aligned}
\quad (40)
$$

By using the monotonicity of $\mathcal{T}$, we have

$$\langle (w^{t+1} - \bar{w}^{t+1}) - (w^t - \bar{w}^t), \bar{w}^{t+1} - \bar{w}^t\rangle_{\mathcal{M}} \geq 0.$$

Hence,

$$\|w^{t+1} - \bar{w}^{t+1}\|_{\mathcal{M}}^2 \leq \|w^t - \bar{w}^t\|_{\mathcal{M}}^2,$$

which together with (38) implies

$$\|w^k - \bar{w}^k\|_{\mathcal{M}}^2 \leq \frac{1}{\rho(2-\rho)(k+1)}\|w^0 - w^*\|_{\mathcal{M}}^2, \ \forall k \geq 0.$$

This completes the proof. $\qquad\square$

To analyze the iteration complexity of Algorithm 2, we begin by considering the residual mapping associated with the KKT system (5), as introduced in Han et al. (2018):

$$\mathcal{R}(w) = \begin{pmatrix} y - \mathrm{Prox}_{f_1}(y - B_1^* x) \\ z - \mathrm{Prox}_{f_2}(z - B_2^* x) \\ c - B_1 y - B_2 z \end{pmatrix}, \quad \forall w = (y, z, x) \in \mathbb{W}. \quad (41)$$

It is clear that $w^* = (y^*, z^*, x^*)$ satisfies the KKT system (5) if and only if $\mathcal{R}(w^*) = 0$. Now, we are ready to present the non-ergodic iteration complexity for Algorithm 2.

**Theorem C.1.** *Suppose that Assumptions 2.1 and 2.2 hold. Let $\{(\bar{y}^k, \bar{z}^k, \bar{x}^k)\}$ be the sequence generated by Algorithm 2 with $\rho \in (0, 2)$, and let $w^* = (y^*, z^*, x^*)$ be the limit point of the sequence $\{(\bar{y}^k, \bar{z}^k, \bar{x}^k)\}$ and $R_0 = \|w^0 - w^*\|_{\mathcal{M}}$. For all $k \geq 0$, we have the following iteration complexity bounds:*

$$\|\mathcal{R}(\bar{w}^k)\| \leq \left( \frac{\sigma\|B_1^*\| + 1}{\sqrt{\sigma}} + \|\sqrt{\mathcal{T}_2}\| + \|\sqrt{\mathcal{T}_1}\| \right) \frac{R_0}{\sqrt{\rho(2-\rho)(k+1)}} \quad (42)$$

*and*

$$\left( \frac{-1}{\sqrt{\sigma}}\|x^*\| \right) \frac{R_0}{\sqrt{\rho(2-\rho)(k+1)}} \leq h(\bar{y}^k, \bar{z}^k) \leq \left( 3R_0 + \frac{1}{\sqrt{\sigma}}\|x^*\| \right) \frac{R_0}{\sqrt{\rho(2-\rho)(k+1)}}.$$

*Furthermore, as $k \to \infty$,*

$$\|\mathcal{R}(\bar{w}^k)\| \leq \left( \frac{\sigma\|B_1^*\| + 1}{\sqrt{\sigma}} + \|\sqrt{\mathcal{T}_2}\| + \|\sqrt{\mathcal{T}_1}\| \right) o(\frac{1}{\sqrt{k+1}}),$$

*and $|h(\bar{y}^k, \bar{z}^k)| = o(\frac{1}{\sqrt{k+1}})$.*

*Proof.* We first estimate the convergence rate of $\mathcal{R}(\bar{w}^k)$ for any $k \geq 0$. According to Proposition C.1, we have

$$\|w^k - \bar{w}^k\|_{\mathcal{M}}^2 \leq \frac{R_0^2}{\rho(2-\rho)(k+1)}, \quad \forall k \geq 0.$$

By the definition of $\mathcal{M}$ in (7), this can be rewritten as

$$\|\bar{y}^k - y^k\|_{\mathcal{T}_1}^2 + \frac{1}{\sigma}\|\sigma B_1(\bar{y}^k - y^k) + (\bar{x}^k - x^k)\|^2 + \|\bar{z}^k - z^k\|_{\mathcal{T}_2}^2 \leq \frac{R_0^2}{\rho(2-\rho)(k+1)}, \forall k \geq 0. \quad (43)$$

Due to Step 2 in Algorithm 2, we can deduce that for any $k \geq 0$,

$$\begin{aligned}\|\sigma B_1(\bar{y}^k - y^k) + (\bar{x}^k - x^k)\| &= \|\sigma B_1(\bar{y}^k - y^k) + \sigma(B_1 y^k + B_2 \bar{z}^k - c)\| \\ &= \sigma\|B_1\bar{y}^k + B_2\bar{z}^k - c\|,\end{aligned}$$

which together with (43) yields that

$$\|B_1\bar{y}^k + B_2\bar{z}^k - c\| \leq \frac{R_0}{\sqrt{\sigma\rho(2-\rho)(k+1)}}, \quad \forall k \geq 0. \quad (44)$$

Moreover, from the optimality conditions of the subproblems in Algorithm 2, we have for any $k \geq 0$,

$$\begin{cases} \bar{z}^k = \mathrm{Prox}_{f_2}(\bar{z}^k - B_2^*\bar{x}^k - \mathcal{T}_2(\bar{z}^k - z^k)), \\ \bar{y}^k = \mathrm{Prox}_{f_1}(\bar{y}^k - B_1^*(\bar{x}^k + \sigma(B_1\bar{y}^k + B_2\bar{z}^k - c)) - \mathcal{T}_1(\bar{y}^k - y^k)), \end{cases} \quad (45)$$

which together with (43) yields that for any $k \geq 0$,

$$\begin{aligned} &\|\bar{z}^k - \mathrm{Prox}_{f_2}(\bar{z}^k - B_2^*\bar{x}^k)\| \\ =\ & \|\mathrm{Prox}_{f_2}(\bar{z}^k - B_2^*\bar{x}^k - \mathcal{T}_2(\bar{z}^k - z^k)) - \mathrm{Prox}_{f_2}(\bar{z}^k - B_2^*\bar{x}^k)\| \\ \leq\ & \|\mathcal{T}_2(\bar{z}^k - z^k)\| \\ \leq\ & \|\sqrt{\mathcal{T}_2}\|\|\bar{z}^k - z^k\|_{\mathcal{T}_2} \\ \leq\ & \|\sqrt{\mathcal{T}_2}\|\frac{R_0}{\sqrt{\rho(2-\rho)(k+1)}}. \end{aligned} \quad (46)$$

Similarly, from (43), (44), and (45), we also have for any $k \geq 0$,

$$\begin{aligned} &\|\bar{y}^k - \mathrm{Prox}_{f_1}(\bar{y}^k - B_1^*\bar{x}^k)\| \\ \leq\ & \|B_1^*\sigma(B_1\bar{y}^k + B_2\bar{z}^k - c) + \mathcal{T}_1(\bar{y}^k - y^k)\| \\ \leq\ & \sigma\|B_1^*\|\|B_1\bar{y}^k + B_2\bar{z}^k - c\| + \|\mathcal{T}_1(\bar{y}^k - y^k)\| \\ \leq\ & (\sqrt{\sigma}\|B_1^*\| + \|\sqrt{\mathcal{T}_1}\|)\frac{R_0}{\sqrt{\rho(2-\rho)(k+1)}}. \end{aligned} \quad (47)$$

Therefore, by (44), (46), and (47), we can obtain that for any $k \geq 0$,

$$\begin{aligned} \|\mathcal{R}(\bar{w}^k)\| &\leq \sqrt{\left(\frac{1}{\sigma} + \|\sqrt{\mathcal{T}_2}\|^2 + (\sqrt{\sigma}\|B_1^*\| + \|\sqrt{\mathcal{T}_1}\|)^2\right)}\frac{R_0}{\sqrt{\rho(2-\rho)(k+1)}} \\ &\leq \left(\frac{\sigma\|B_1^*\|+1}{\sqrt{\sigma}} + \|\sqrt{\mathcal{T}_2}\| + \|\sqrt{\mathcal{T}_1}\|\right)\frac{R_0}{\sqrt{\rho(2-\rho)(k+1)}}. \end{aligned} \quad (48)$$

Now, we estimate the complexity result concerning the objective error. For the lower bound of the objective error, from Sun et al. (2024, Lemma 3.6) and (44), we have for all $k \geq 0$,

$$\begin{aligned} h(\bar{y}^k, \bar{z}^k) &\geq \langle B_1\bar{y}^k + B_2\bar{z}^k - c, -x^*\rangle \\ &\geq -\|x^*\|\|B_1\bar{y}^k + B_2\bar{z}^k - c\| \\ &\geq -\frac{R_0\|x^*\|}{\sqrt{\sigma\rho(2-\rho)(k+1)}}. \end{aligned}$$

On the other hand, from the (22), we can obtain

$$\|w^{k+1} - w^*\|_{\mathcal{M}} \leq \|w^k - w^*\|_{\mathcal{M}} \leq \dots \leq R_0, \quad \forall k \geq 0.$$

It follows from the $\mathcal{M}$-nonexpansiveness of $\widehat{\mathcal{T}}$ by Sun et al. (2024, Proposition 2.3) that

$$\|\bar{w}^k - w^*\|_{\mathcal{M}} = \|\widehat{\mathcal{T}}w^k - w^*\|_{\mathcal{M}} \leq \|w^k - w^*\|_{\mathcal{M}} \leq R_0, \quad \forall k \geq 0,$$

which implies

$$\|\bar{y}^k - y^*\|_{\mathcal{T}_1}^2 + \frac{1}{\sigma}\|\sigma B_1(\bar{y}^k - y^*) + (\bar{x}^k - x^*)\|^2 + \|\bar{z}^k - z^*\|_{\mathcal{T}_2}^2 \le R_0^2, \quad \forall k \ge 0.$$

This inequality together with (30), (43), and (44) yields that for all $k \ge 0$,

$$\begin{aligned} h(\bar{y}^k, \bar{z}^k) & \le (\|\sigma B_1(\bar{y}^k - y^*) + (\bar{x}^k - x^*)\| + \|x^*\|)\|B_1\bar{y}^k + B_2\bar{z}^k - c\| \\ & \quad + \|y^* - \bar{y}^k\|_{\mathcal{T}_1}\|\bar{y}^k - y^k\|_{\mathcal{T}_1} + \|z^* - \bar{z}^k\|_{\mathcal{T}_2}\|\bar{z}^k - z^k\|_{\mathcal{T}_2} \\ & \le (\sqrt{\sigma}R_0 + \|x^*\|)\frac{R_0}{\sqrt{\sigma\rho(2-\rho)(k+1)}} + \frac{2R_0^2}{\sqrt{\rho(2-\rho)(k+1)}} \\ & = \left(3R_0 + \frac{1}{\sqrt{\sigma}}\|x^*\|\right)\frac{R_0}{\sqrt{\rho(2-\rho)(k+1)}}. \end{aligned}$$

We now establish the complexity results as $k \to \infty$. According to Proposition C.1, we have

$$\left\|w^k - \bar{w}^k\right\|_{\mathcal{M}}^2 = o\left(\frac{1}{k+1}\right).$$

Following a similar approach as in the previous proof, we obtain

$$\|\mathcal{R}(\bar{w}^k)\| \le \left(\frac{\sigma\|B_1^*\| + 1}{\sqrt{\sigma}} + \|\sqrt{\mathcal{T}_2}\| + \|\sqrt{\mathcal{T}_1}\|\right)o\left(\frac{1}{\sqrt{k+1}}\right),$$

and

$$|h(\bar{y}^k, \bar{z}^k)| = o\left(\frac{1}{\sqrt{k+1}}\right).$$

This completes the proof. $\qquad\square$

## D  COMPARISON OF THE ITERATION COMPLEXITY

In addition to the pADMM in Algorithm 2, another widely used variant of the semi-proximal ADMM for solving COPs, as introduced by Fazel et al. (2013), is presented in Algorithm 4.

---

**Algorithm 4** A semi-proximal ADMM (sPADMM) for solving COP (3)

---

1: Input: Let $\mathcal{T}_1$ and $\mathcal{T}_2$ be two self-adjoint, positive semidefinite operators on $\mathbb{Y}$ and $\mathbb{Z}$, respectively. Select an initial point $(y^0, z^0, x^0) \in \text{dom}(f_1) \times \text{dom}(f_2) \times \mathbb{X}$. Set the parameters $\sigma > 0$ and $\tau \in (0, \frac{1+\sqrt{5}}{2})$.

2: **for** $k = 0, 1, ...,$ **do**

3:     Step 1. $y^{k+1} = \underset{y \in \mathbb{Y}}{\arg\min}\left\{L_\sigma\left(y, z^k; x^k\right) + \frac{1}{2}\|y - y^k\|_{\mathcal{T}_1}^2\right\}$;

4:     Step 2. $z^{k+1} = \underset{z \in \mathbb{Z}}{\arg\min}\left\{L_\sigma\left(y^{k+1}, z; x^k\right) + \frac{1}{2}\|z - z^k\|_{\mathcal{T}_2}^2\right\}$;

5:     Step 3. $x^{k+1} = x^k + \tau\sigma(B_1y^{k+1} + B_2z^{k+1} - c)$;

---

Unlike Algorithm 4, the pADMM in Algorithm 2 can be reformulated as a dPPM, facilitating the analysis of its ergodic convergence properties. We summarize some iteration complexity results of these two algorithms in Table 2.

Table 2: The iteration complexity result of pADMM/sPADMM

| Paper | Algorithm | Proximal operators | $\rho$ | $\tau$ | Feasibility violation | Objective error | KKT residual | Type |
|---|---|---|---|---|---|---|---|---|
| Davis & Yin (2016) | GADMM | $\mathcal{T}_1 = 0, \mathcal{T}_2 = 0$ | $(0, 2)$ | $1$ | $o(1/\sqrt{k})$ | $o(1/\sqrt{k})$ | - | non-ergodic |
| Cui et al. (2016) | (majorized) sPADMM | $\mathcal{T}_1 \succeq 0, \mathcal{T}_2 \succeq 0^{\text{a}}$ | $1$ | $(0, \frac{1+\sqrt{5}}{2})$ | $o(1/\sqrt{k})$ | - | $o(1/\sqrt{k})$ | non-ergodic |
| Ours | pADMM | $\mathcal{T}_1 \succeq 0, \mathcal{T}_2 \succeq 0$ | $(0, 2)$ | $1$ | $o(1/\sqrt{k})$ | $o(1/\sqrt{k})$ | $o(1/\sqrt{k})$ | non-ergodic |
| Monteiro & Svaiter (2013) | ADMM | $\mathcal{T}_1 = 0, \mathcal{T}_2 = 0$ | $1$ | $1$ | $O(1/k)$ | - | $O_\varepsilon(1/k)^{\text{b}}$ | ergodic |
| Davis & Yin (2016) | GADMM | $\mathcal{T}_1 = 0, \mathcal{T}_2 = 0$ | $(0, 2]$ | $1$ | $O(1/k)$ | $O(1/k)$ | - | ergodic |
| Cui et al. (2016) | (majorized) sPADMM | $\mathcal{T}_1 \succeq 0, \mathcal{T}_2 \succeq 0$ | $1$ | $(0, \frac{1+\sqrt{5}}{2})$ | $O(1/k)$ | $O(1/k)$ | - | ergodic |
| Adona et al. (2019) | GADMM | $\mathcal{T}_1 \succeq 0, \mathcal{T}_2 \succeq 0$ | $(0, 2]$ | $1$ | $O(1/k)$ | - | $O_\varepsilon(1/k)$ | ergodic |
| Shen & Pan (2016) | sPADMM | $\mathcal{T}_1 \succeq 0, \mathcal{T}_2 \succeq 0$ | $1$ | $(0, \frac{1+\sqrt{5}}{2})$ | $O(1/k)$ | - | $O_\varepsilon(1/k)$ | ergodic |
| Ours | pADMM | $\mathcal{T}_1 \succeq 0, \mathcal{T}_2 \succeq 0$ | $(0, 2]$ | $1$ | $O(1/k)$ | $O(1/k)$ | $O_\varepsilon(1/k)$ | ergodic |

a $\mathcal{T}_1 \succeq 0$ denotes that $\mathcal{T}_1$ is positive semi-definite.
b $O_\varepsilon(1/k)$ of the KKT residual: an $O(1/k)$ iteration complexity of the KKT residual based on $\varepsilon$ subdifferntin in (13).

**Remark D.1.** *Note that pADMM with $\mathcal{T}_1 = 0$, $\mathcal{T}_2 = 0$, and $\rho = 2$ is equivalent to GADMM. In comparison to the results of Davis & Yin (2016), our work not only establishes both ergodic and non-ergodic iteration complexities for the KKT residual—an aspect not addressed in Davis & Yin (2016)—but also extends the analysis to incorporate general $\mathcal{T}_1$ and $\mathcal{T}_2$. This generalization is particularly important, as suitable choices of $\mathcal{T}_1$ and $\mathcal{T}_2$ can simplify the solution of subproblems in solving key convex optimization problems, such as general LP. Furthermore, the GADMM algorithm with semi-proximal terms studied in (Adona et al., 2019) differs from pADMM due to the way the proximal terms are incorporated.*

To further discuss the relationship between the KKT residual based on $\varepsilon$-subdifferential in (13) and the KKT residual defined in (41), we first present the following lemma to highlight the difference between $\partial_\varepsilon f(\cdot)$ and $\mathrm{Prox}_f(\cdot)$ for a proper closed convex function $f$.

**Lemma D.1.** *Let $f : \mathbb{X} \to (-\infty, +\infty]$ be a proper closed convex function, and let $\bar{x} \in \mathrm{dom}(f)$. Given $\varepsilon \geq 0$, if $v \in \partial_\varepsilon f(\bar{x})$, then*

$$\|\bar{x} - \mathrm{Prox}_f(\bar{x} + v)\|^2 \leq \varepsilon. \tag{49}$$

*Proof.* For notational convenience, denote $\mathrm{Prox}_f(\bar{x} + v)$ by $\widetilde{x}$. According to the definition of $\mathrm{Prox}_f(\bar{x} + v)$, we have

$$0 \in \partial f(\widetilde{x}) + (\widetilde{x} - (\bar{x} + v)),$$

which implies

$$f(x) \geq f(\widetilde{x}) + \langle \bar{x} + v - \widetilde{x}, x - \widetilde{x} \rangle, \forall x \in \mathbb{X}.$$

It follows that

$$\begin{aligned} f(\bar{x}) \; &\geq f(\widetilde{x}) + \langle \bar{x} + v - \widetilde{x}, \bar{x} - \widetilde{x} \rangle, \\ &= f(\widetilde{x}) + \|\bar{x} - \widetilde{x}\|^2 + \langle v, \bar{x} - \widetilde{x} \rangle. \end{aligned} \tag{50}$$

In addition, according to assumption that $v \in \partial_\varepsilon f(\bar{x})$, we have

$$f(\widetilde{x}) \geq f(\bar{x}) + \langle v, \widetilde{x} - \bar{x} \rangle - \varepsilon. \tag{51}$$

Summing (50) and (51), we have

$$\|\widetilde{x} - \bar{x}\|^2 \leq \varepsilon.$$

This completes the proof. $\qquad\square$

**Remark D.2.** *The following strongly convex quadratic function demonstrates that the inequality in Lemma D.1 is tight up to a constant factor of 2:*

$$f(x) := \frac{1}{2}\langle x, x \rangle + \langle b, x \rangle,$$

where $b \in \mathbb{R}^n$. As shown in Hiriart-Urruty & Lemaréchal (1993, Example 1.2.2), it follows that

$$\partial_\varepsilon f(x) = \left\{ x + b + u : \frac{1}{2}\langle u, u \rangle \leq \varepsilon \right\}.$$

Consider $\bar{x} \in \operatorname{dom}(f)$. For any $\varepsilon \geq 0$, we take the $\varepsilon$-subgradient $v$ of $f(\cdot)$ at $\bar{x}$ as follows:

$$v = \bar{x} + b + u,$$

where $\frac{1}{2}\|u\|^2 = \varepsilon$. Through direct calculation, we obtain

$$\|\bar{x} - \operatorname{Prox}_f(\bar{x} + v)\|^2 = \|\bar{x} - \bar{x} - \frac{u}{2}\|^2 = \frac{1}{4}\|u\|^2 = \frac{1}{2}\varepsilon.$$

This demonstrates that the inequality in (49) is tight, up to the constant factor of 2.

Now, we are ready to give the iteration complexity of the ergodic sequence $\{\bar{w}_a^k\}$ generated by Algorithm 2 in the following theorem.

**Theorem D.1.** *Suppose that Assumptions 2.1 and 2.2 hold. Let $\{(\bar{y}_a^k, \bar{z}_a^k, \bar{x}_a^k)\}$ be the sequence generated by Algorithm 2 with $\rho \in (0, 2]$. Let $w^* = (y^*, z^*, x^*)$ be a solution to the KKT system (5), and $R_0 = \|w^0 - w^*\|_{\mathcal{M}}$. For all $k \geq 0$, we have the following iteration complexity bound:*

$$\|\mathcal{R}(\bar{w}_a^k)\| \leq \frac{2R_0}{\sqrt{2\rho(k+1)}} + \left( \frac{\sigma\|B_1^*\| + 1}{\sqrt{\sigma}} + \|\sqrt{\mathcal{T}_2}\| + \|\sqrt{\mathcal{T}_1}\| \right) \frac{2R_0}{\rho(k+1)}. \tag{52}$$

*Proof.* According to the Lemma 2.1, we have for $k \geq 0$,

$$\begin{cases} -B_2^* \bar{x}_a^k - \mathcal{T}_2(\bar{z}_a^k - z_a^k) \in \partial_{\bar{\varepsilon}_z^k} f_2(\bar{z}_a^k), \\ -B_1^* (\bar{x}_a^k + \sigma(B_1 \bar{y}_a^k + B_2 \bar{z}_a^k - c)) - \mathcal{T}_1(\bar{y}_a^k - y_a^k) \in \partial_{\bar{\varepsilon}_y^k} f_1(\bar{y}_a^k), \end{cases}$$

which together with Lemma D.1 implies that

$$\begin{cases} \|\bar{z}_a^k - \operatorname{Prox}_{f_2}(\bar{z}_a^k - B_2^* \bar{x}_a^k - \mathcal{T}_2(\bar{z}_a^k - z_a^k))\|^2 \leq \bar{\varepsilon}_z^k, \\ \|\bar{y}_a^k - \operatorname{Prox}_{f_1}(\bar{y}_a^k - B_1^*(\bar{x}_a^k + \sigma(B_1 \bar{y}_a^k + B_2 \bar{z}_a^k - c)) - \mathcal{T}_1(\bar{y}_a^k - y_a^k)\|^2 \leq \bar{\varepsilon}_y^k. \end{cases} \tag{53}$$

It follows that for any $k \geq 0$,

$$\begin{aligned} & \|\bar{z}_a^k - \operatorname{Prox}_{f_2}(\bar{z}_a^k - B_2^* \bar{x}_a^k)\| \\ \leq \ & \|\bar{z}_a^k - \operatorname{Prox}_{f_2}(\bar{z}_a^k - B_2^* \bar{x}_a^k - \mathcal{T}_2(\bar{z}_a^k - z_a^k))\| \\ & + \|\operatorname{Prox}_{f_2}(\bar{z}_a^k - B_2^* \bar{x}_a^k - \mathcal{T}_2(\bar{z}_a^k - z_a^k)) - \operatorname{Prox}_{f_2}(\bar{z}_a^k - B_2^* \bar{x}_a^k)\| \\ \leq \ & \sqrt{\bar{\varepsilon}_z^k} + \|\mathcal{T}_2(\bar{z}_a^k - z_a^k)\|. \end{aligned} \tag{54}$$

Hence, from (12) and (26), we can obtain

$$\|\bar{z}_a^k - \operatorname{Prox}_{f_2}(\bar{z}_a^k - B_2^* \bar{x}_a^k)\| \leq \frac{R_0}{\sqrt{2\rho(k+1)}} + \|\sqrt{\mathcal{T}_2}\| \frac{2R_0}{\rho(k+1)}. \tag{55}$$

Similarly, from (12), (26), (27), and (53), we also have for any $k \geq 0$,

$$\begin{aligned} & \|\bar{y}_a^k - \operatorname{Prox}_{f_1}(\bar{y}_a^k - B_1^* \bar{x}_a^k)\| \\ \leq \ & \sqrt{\bar{\varepsilon}_y^k + \|B_1^* \sigma(B_1 \bar{y}_a^k + B_2 \bar{z}_a^k - c) + \mathcal{T}_1(\bar{y}_a^k - y_a^k)\|} \\ \leq \ & \sqrt{\bar{\varepsilon}_y^k} + \sigma\|B_1^*\|\|B_1 \bar{y}_a^k + B_2 \bar{z}_a^k - c\| + \|\mathcal{T}_1(\bar{y}_a^k - y_a^k)\| \\ \leq \ & \frac{R_0}{\sqrt{2\rho(k+1)}} + (\sqrt{\sigma}\|B_1^*\| + \|\sqrt{\mathcal{T}_1}\|) \frac{2R_0}{\rho(k+1)}. \end{aligned} \tag{56}$$

Therefore, by (27), (55), and (56), we can obtain that for any $k \geq 0$,

$$\begin{aligned} \|\mathcal{R}(\bar{w}_a^k)\| \ & \leq \frac{2R_0}{\sqrt{2\rho(k+1)}} + (\frac{1}{\sqrt{\sigma}} + \sqrt{\sigma}\|B_1^*\| + \|\sqrt{\mathcal{T}_1}\| + \|\sqrt{\mathcal{T}_2}\|) \frac{2R_0}{\rho(k+1)} \\ & \leq \frac{2R_0}{\sqrt{2\rho(k+1)}} + \left( \frac{\sigma\|B_1^*\| + 1}{\sqrt{\sigma}} + \|\sqrt{\mathcal{T}_2}\| + \|\sqrt{\mathcal{T}_1}\| \right) \frac{2R_0}{\rho(k+1)}. \end{aligned} \tag{57}$$

This completes the proof. $\qquad\square$

## E  UPDATE RULE FOR $\sigma$ WITH SAFEGUARDS

Since the approximations $\Delta_x$ and $\Delta_y$ may significantly deviate from their true values, we update $\sigma$ using formula (21) only when the following conditions are met; otherwise, we reset $\sigma$ to 1:

1. $\Delta_x$ and $\Delta_y$ are within the range:
$$\Delta_x, \Delta_y \in (10^{-16}, 10^{12}); \tag{58}$$

2. The ratio of relative primal and dual infeasibility errors is within acceptable bounds:
$$\frac{\text{error}_d}{\text{error}_p} \in (10^{-8}, 10^8), \tag{59}$$

where
$$\text{error}_p := \frac{\|\Pi_D(b - A\bar{x}_a^{r,\tau_r})\|}{1 + \|b\|} \quad \text{and} \quad \text{error}_d := \frac{\|c - A^*\bar{y}_a^{r,\tau_r} - \bar{z}_a^{r,\tau_r}\|}{1 + \|c\|}.$$

In summary, the update rule for $\sigma$ is presented in Algorithm 5.

---

**Algorithm 5 SigmaUpdate**

---

1: **Input:** $(\bar{w}^{r,\tau_r}, w^{r,0}, \mathcal{S}_1, A)$.
2: Calculate $\Delta_x$ and $\Delta_y$ defined in (20);
3: **if** conditions (58) and (59) are satisfied **then**
4: $\qquad \sigma_{r+1} = \dfrac{\Delta_x}{\Delta_y}$;
5: **else**
6: $\qquad \sigma_{r+1} = 1$;
7: **Output:** $\sigma_{r+1}$.

---

## F  EXPERIMENTAL SETUP

**Benchmark datasets.** Our benchmark datasets include Mittelmann's LP benchmark set and LP relaxations of instances from the MIPLIB 2017 collection. We test the algorithms on 49 publicly available instances from Mittelmann's LP benchmark. From the MIPLIB 2017 collection, we select 383 instances following the criteria in Lu & Yang (2023). Of these, two are reported as unbounded, and one is solved by Gurobi's presolve (Gurobi Optimization, LLC, 2024), leaving 380 instances for testing.

**Software and computing environment.** EPR-LP and EDR-LP are implemented in Julia (Bezanson et al., 2017). For a fair comparison, the infeasibility detection of cuPDLP (Lu & Yang, 2023) is disabled. All algorithms are tested on an NVIDIA A100-SXM4-80GB GPU with CUDA 12.3.

**Presolve and preconditioning.** We compare all algorithms across all datasets, both with and without presolve (using Gurobi 11.0.3, academic license). Before running EPR-LP and EDR-LP, all problems are preconditioned for numerical stability, involving 10 steps of Ruiz scaling (Ruiz, 2001), followed by bidiagonal preconditioning as described in Pock & Chambolle (2011) with $\alpha = 1$. Finally, the vectors $b$ and $c$ are normalized by $\|b\| + 1$ and $\|c\| + 1$, respectively. cuPDLP (Lu & Yang, 2023) uses its default settings.

**Initialization and parameter setting.** The initial points of EPR-LP and EDR-LP are the origin. We set the penalty parameter $\sigma_0 = 1$. After preconditioning, we estimate $\lambda_1(AA^*)$ by the power method (Golub & Van Loan, 2013).

**Termination criteria.** We check the stopping criteria for the sequence $\{\bar{w}_a^{r,t}\}$ for EPR-LP and EDR-LP. The feasibilities $\bar{x}_a^{r,t} \in C$ and $\bar{y}_a^{r,t} \in D$ are satisfied for any $r \geq 0$, and $t \geq 1$. We terminate the algorithms when the following stopping criteria (used in Applegate et al. (2021); Lu & Yang (2023); Lu et al. (2023)) are satisfied for the tolerance $\varepsilon \in (0, \infty)$:

$$|\langle b, y \rangle - \delta_C^*(-z) - \langle c, x \rangle| \leq \varepsilon \left( 1 + |\langle b, y \rangle - \delta_C^*(-z)| + |\langle c, x \rangle| \right),$$
$$\|\Pi_D(b - Ax)\| \leq \varepsilon \left( 1 + \|b\| \right),$$
$$\|c - A^*y - z\| \leq \varepsilon \left( 1 + \|c\| \right).$$

We test all algorithms with $\varepsilon = 10^{-8}$ for all the datasets.

**Time limit.** We set a time limit of 15,000 seconds for Mittelmann's benchmark dataset. For the LP relaxations of MIP problems, the time limit is 3,600 seconds if the number of nonzero elements in $A$ is less than 10 million; otherwise, the limit is 18,000 seconds.

## G SUPPLEMENTARY EXPERIMENTS

### G.1 SPARSITY OF THE SOLUTION

In this subsection, we use the optimal transport (OT) problem (Monge, 1781; Kantorovich, 1942) as an example to demonstrate that EPR-LP, combined with a restart strategy, effectively preserves the sparsity of the solution. For the experiment, we calculate the optimal transport mapping between two 64x64 pixel images selected from the shape category in the DOTmark dataset (Schrieber et al., 2016). The sparsity of the variable $x$ is shown in Figure 4. It can be observed that, with the restart strategy, EPR-LP quickly maintains the sparsity of $x$ similar to the nonergodic version. Consequently, the sparsity has little effect on the computational efficiency of EPR-LP.

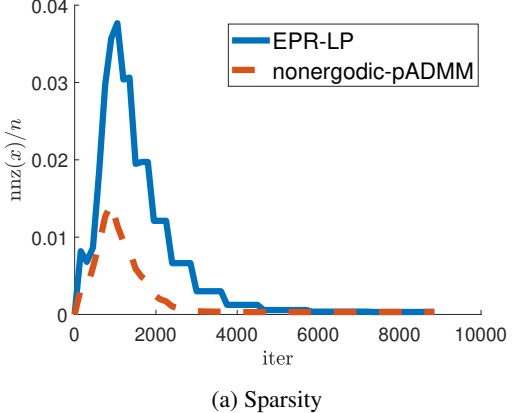

(a) Sparsity

| Algorithm | Iterations | Time (s) |
|---|---|---|
| Nonergodic pADMM | 8,529,300 | 19,514.3 |
| EPR-LP | 220,050 | 514.9 |

(b) Iteration number and time

Figure 4: (a) The sparsity of $x$. (b) Comparison of iteration count and runtime between nonergodic pADMM ($\rho = 1.6$) and EPR-LP with a tolerance of $10^{-8}$.

