# OpenReview forum: "On the ergodic convergence properties of the Peaceman-Rachford method and their applications in solving linear programming"
_ICLR.cc/2025/Conference — ICLR 2025 Conference Withdrawn Submission_

### Official Review · Reviewer_vXig · 2024-11-02

**Soundness:** 2
**Presentation:** 3
**Contribution:** 2
**Rating:** 5
**Confidence:** 3

**Summary:**

In this paper, the authors proved the ergodic convergence of the PR method with semi-proximal terms for solving convex optimization problems. We established the ergodic iteration complexity of $\mathcal{O}(1/k)$. Building on these results, the authors developed the solver EPR-LP for solving large-scale LP problems, which incorporates adaptive restart and penalty parameter updates. The authors also conduct numerical experiments and showed empirical results that are consistent with the theoretical analysis.

**Strengths:**

In this submission, the authors established the ergodic convergence of the dPPM and proved the ergodic convergence of the pADMM through reformulating the pADMM as adPPM. They also proposed the solver EPR-LP, an implementation of the PR method that utilizes the ergodic sequence to solve LPs, incorporating an adaptive restart strategy and dynamic penalty parameter updates. The authors provided all the detailed mathematical proof and analysis as well as extensive numerical experiments on LP benchmark datasets. The structure of this submission is clear with high quality presentations.

**Weaknesses:**

The only weakness of this paper is that the authors should present and proof the convergence analysis for the proposed algorithm 3: EPR-LP: A Peaceman-Rachford method using ergodic sequence. Although the authors presented the details for the restart strategy and the update rule for the $\sigma$, the authors should present the convergence results of the proposed method similar to Theorem 2.1 and Theorem 2.2. If the authors could supplement the theoretical analysis for the proposed algorithm, this submission deserves the acceptance of the publishment.

**Questions:**

Could the authors provide some convergence theorems or results for the proposed algorithm 3?

---

> ### Author Response · Authors · 2024-11-27
>
> We sincerely appreciate your positive feedback and thoughtful suggestions! To address your concerns, we provide detailed responses below.
>
> ---
>
> ### **Response to the weaknesses**
>
> 1. **Weakness 1: Could the authors provide some convergence theorems or results for the proposed EPR-LP (Algorithm 3)?**
>    **Response:**
>    Thank you for your question. Our current theoretical results are in the settings of a fixed penalty parameter without restart. We include these heuristics in EPR-LP because they can improve the practical performance of the algorithms. In the implementation of the algorithm, we usually set a maximum number of restarts. Therefore, the convergence guarantee in Corollary 2.1 still holds under practical settings with parameter tuning and restart as a direct consequence. However, the iteration complexity now only holds for iterations between two restarts. Here, we would like to further highlight the key contributions of this paper:
>
> - This paper presents significant progress in understanding the convergence properties of the PR method by proving the convergence of its ergodic sequence. More importantly, numerical results demonstrate the superior performance of the PR method’s ergodic sequence compared to that of the DR method. This finding resolves the long-standing puzzle of why the non-ergodic sequence of the PR method outperforms that of the DR method when the PR method does converge.  In particular, we study the convergence properties for a more general semi-proximal PR algorithm, which includes the PR method as a special case. Here, we highlight the motivation and importance of the extra semi-proximal terms in the considered semi-proximal PR algorithm: for solving large-scale convex optimization problems like LP problems, appropriate selection of the proximal operators $ \mathcal{T}_1 $ and $ \mathcal{T}_2 $ are necessary for solving the subproblems efficiently.
>
> - We have established the $ O(1/k) $ ergodic iteration complexity for the semi-proximal PR method, with respect to feasibility violation, objective error, and KKT-residual based on $ \varepsilon $-subdifferential. A key challenge arises from the fact that the concerned function may not be sub-differentiable at the ergodic point, even if it is sub-differentiable at all the points in the non-ergodic sequence. We leverage the $ \varepsilon $-subgradient to overcome this challenge.
>
> We also want to highlight that the preliminary numerical results shown in this paper have demonstrated the superior numerical performance of EPR-LP for solving general large-scale LPs, compared to the award-winning solver cuPDLP [Applegate et al. (2021), Applegate et al. (2023), Lu and Yang (2023)].
>
> ---
> ### **References**
>
> 1. Applegate, David, Mateo Díaz, Oliver Hinder, Haihao Lu, Miles Lubin, Brendan O'Donoghue, and Warren Schudy. "Practical large-scale linear programming using primal-dual hybrid gradient." *Advances in Neural Information Processing Systems* 34 (2021): 20243-20257.
> 2. Applegate, David, Oliver Hinder, Haihao Lu, and Miles Lubin. "Faster first-order primal-dual methods for linear programming using restarts and sharpness." *Mathematical Programming* 201, no. 1 (2023): 133-184.
> 3. Lu, Haihao, and Jinwen Yang. "cuPDLP.jl: A GPU implementation of restarted primal-dual hybrid gradient for linear programming in Julia." *arXiv preprint* arXiv:2311.12180 (2023).
>
> ---
> We hope our response addresses your concerns. We sincerely appreciate your re-evaluation of our paper.

---

### Official Review · Reviewer_7tP9 · 2024-11-03

**Soundness:** 3
**Presentation:** 2
**Contribution:** 3
**Rating:** 6
**Confidence:** 3

**Summary:**

This paper studies the convergence and iteration complexity of the ergodic sequence generated by a fairly general preconditioned semi-proximal ADMM method. As a result, the authors establish the convergence rate of the ergodic sequence generated by the PR method. Motivated by recent developments in first-order algorithms for solving large-scale LPs, they instantiate the algorithm to solve the LP problem and conduct experiments to compare the proposed method with a state-of-the-art method, PDLP.

**Strengths:**

When the quality of intermediate solutions suffices and the dimensionality of the data is large, parallelizable first-order methods are preferred over conventional IPMs and the simplex method. Recently, there has been a surge of investigation into designing highly efficient first-order methods for solving LPs. This work fits well within this line of research and establishes convergence behavior and rates for settings that were previously unknown.

**Weaknesses:**

Overall, I believe the contribution is publishable, and the motivation is plausible. My comments are as follows.

* The proposed EPR-LP uses restart and dynamic penalty tuning. Are these heuristics covered in the convergence analysis?

* In Line 1 of Algorithm 1, compared with Algorithm 2, the variable $z^0$ may not be in the effective domain of the augmented Lagrangian. Could this cause issues for the algorithm, especially when rho is close to one?

* In the paragraph starting on Line 066, the authors mention that without the preconditioner and with $\rho=2$, Algorithm 1 is equivalent to GADMM induced by the PR method. Is this GADMM related to the GADMM of Davis-Yin in Table 2? If so, and if I understand correctly, as shown by the fourth line in Table 2, the ergodic convergence rate for the rho=2 case has already been established in Davis-Yin. I suggest the authors elaborate on the subtle differences between the algorithms referenced in Table 2, as it could be confusing to readers which cases are not covered in the literature and are introduced for the first time in this paper.

* Compared with other non-ergodic results in Table 2, why is the non-ergodic analysis in Appendix B only of Big-O type rather than small-o?

* For the figures in Section 4, what is the definition of the horizontal axis? I think it would be helpful to provide more details on the rationale for this choice.

* Regarding the third bullet on Line 454, I don’t think this observation is very meaningful. First, the bounds given in (13) and (14) represent worst-case behavior, which could be far from the actual data distribution in real-world datasets. Second, it appears that methods EPR-LP and EDR-LP use heuristic techniques, such as restart and penalty tuning, to improve numerical performance, which may not be reflected in (13) and (14).


Minor:
* The meaning of the abbreviation EPR-LP should be explained early before frequently used. Indeed, the first appearance of EPR-LP is at the abstract but only formally defined on Line 330 in page 7.

* The notation $A$ is defined as a matrix in Line 043, but could be used as a linear operator in, e.g., Line 075. This is not actually a problem, but making the terminology consistent would be preferable.

* Algorithm 2 appears two pages before its formal definition.

* Line 379: "Since" should be "since"

**Questions:**

See above.

---

> ### Author Response · Authors · 2024-11-27
>
> We sincerely appreciate your positive feedback and thoughtful suggestions! To address your concerns, we provide detailed responses below.
>
> ---
>
> ### **Response to the weaknesses**
>
> 1. **Weakness 1: The proposed EPR-LP uses restart and dynamic penalty tuning. Are these heuristics covered in the convergence analysis?**
>    **Response:**  Thank you for your question. Our current theoretical results are in the settings of a fixed penalty parameter without restart. We include these heuristics in EPR-LP because they can improve the practical performance of the algorithms. In the implementation of the algorithm, we usually set a maximum number of restarts. Therefore, the convergence guarantee in Corollary 2.1 still holds under the practical settings with parameter tuning and restart as a direct consequence.
>
> ---
>
> 2. **Weakness 2: In Line 1 of Algorithm 1, compared with Algorithm 2, the variable $z^{0}$ may not be in the effective domain of the augmented Lagrangian.**
>    **Response:**  Thank you for your comments. We require $(z^0, y^0, x^0) \in {\rm dom}(f_1) \times {\rm dom}(f_2) \times \mathbb{X}$; otherwise, the augmented Lagrangian function will take the value of $+\infty$. We have revised Algorithm 1 and Algorithm 3 accordingly. Here, we want to mention that finding an initial point $(z^0, y^0, x^0) \in {\rm dom}(f_1) \times {\rm dom}(f_2) \times \mathbb{X}$ is not difficult for most optimization problems.
>
> ---
>
> 3. **Weakness 3: Is this GADMM related to the GADMM of Davis-Yin in Table 2? Elaborate on the subtle differences between the algorithms referenced in Table 2.**
>    **Response:**  Thank you for your questions and suggestions. The pADMM in Algorithm 2, with $\mathcal{T}_1 = 0$ and $\mathcal{T}_2 = 0$, corresponds to the GADMM derived from the PR method (Eckstein & Bertsekas, 1992). Below, we highlight key differences between our results and those in the existing works:
>
>    - The iteration complexity of GADMM ($\mathcal{T}_1 = 0$ and $\mathcal{T}_2 = 0$) established by Davis & Yin (2016) focuses on feasibility violation and objective error. In contrast, our work additionally establishes the iteration complexity for the KKT residual.
>    - We also establish the ergodic convergence rate for pADMM with semi-proximal terms ($\mathcal{T}_1 \succeq 0, \mathcal{T}_2 \succeq 0$), encompassing feasibility violation, objective error, and KKT residual based on $\varepsilon$-subdifferential. For solving large-scale convex optimization problems, such as general LPs, appropriately chosen semi-proximal operators $\mathcal{T}_1$ and $\mathcal{T}_2$ are necessary for solving subproblems efficiently.
>
>    Please refer to the updated Table 2 and Remark E.1 in the revised manuscript for more details.
>
> ---
>
> 4. **Weakness 4: Compared with other non-ergodic results in Table 2, why is the non-ergodic analysis in Appendix B only of Big-O type rather than small-o?**
>    **Response:**  Thank you for your question. We have refined the analysis of the non-ergodic case and obtained an $o(1/\sqrt{k})$ iteration complexity result. Please see Appendix C in the revised manuscript for details.
>
> ---
>
> 5. **Weakness 5: For the figures in Section 4, what is the definition of the horizontal axis?**
>    **Response:**
>    Thank you for your question. In Section 4, we use performance profiles as proposed by Dolan and Moré (2002), which are cited in our paper, to compare the performance of cuPDLP and EPR-LP. Performance profiles are a classical and widely accepted tool for benchmarking optimization software, including LP solvers.
>
>    Disregarding the $\log$ scaling on the horizontal axis, $x$ represents the performance ratio, measuring how much worse a solver performs compared to the best solver for a given problem. Specifically, for a problem $p$:
>    - $t_{p,s}$: The computational cost (e.g., time, iterations) of solver $s$ on problem $p$.
>    - $t_{p,\text{best}} = \min_s t_{p,s}$: The computational cost of the best-performing solver on $p$.
>
>    The **performance ratio** for solver $s$ on problem $p$ is defined as:
>    $$
>    r_{p,s} = \frac{t_{p,s}}{t_{p,\text{best}}}.
>    $$
>
>    - $r_{p,s} = 1$: Solver $s$ is the best solver for problem $p$.
>    - $r_{p,s} > 1$: Solver $s$ is slower than the best solver by a factor of $r_{p,s}$.
>
>    The horizontal axis $x$ represents the upper bound for the performance ratio $r_{p,s}$:
>    - At $x = 1$, only the best-performing solvers ($r_{p,s} = 1$) are considered.
>    - At $x = 2$, solvers that are at most twice as slow as the best solver ($r_{p,s} \leq 2$) are included.
>    - As $x$ increases, the comparison becomes more inclusive, eventually considering all solvers for all problems.
>
>    For further details, please refer to the paper by Dolan and Moré (2002).
>
> ---

---

> > ### Author Response · Authors · 2024-11-27
> >
> > ### **Response to the weaknesses**
> >
> > 6. **Weakness 6: The performance of EPR-LP and EDR-LP may not be reflected in the bounds given in (13) and (14), as these methods incorporate heuristic techniques, whereas the bounds represent worst-case behavior.**
> >    **Response:**  Thank you for your question. We agree that the bounds in (13) and (14) are in the sense of worst-case, which may not fully reflect the practical performance of the algorithms. However, since both EPR-LP and EDR-LP employ the same heuristic techniques, the observed numerical performance differences can be reasonably predicted by their iteration complexity bounds. We have revised our manuscript for more detailed clarifications in Section 4.1.
> >
> > ---
> >
> > 7. **Weakness 7: The meaning of the abbreviation EPR-LP should be explained early before being frequently used.**
> >    **Response:**  Thank you for your suggestions. We first introduce the full name of EPR-LP in Line 143 of our manuscript. We have now highlighted it in bold font in the revised version.
> >
> > ---
> >
> > 8. **Weakness 8: The notation $A$ is defined as a matrix in Line 43 but could be used as a linear operator in, e.g., Line 75.**
> >    **Response:**  Thank you for your comment. We kindly ask for some flexibility with the abuse of this notation. We have added a remark in Line 44 in the revised manuscript.
> >
> > ---
> >
> > 9. **Weakness 9: Algorithm 2 appears two pages before its formal definition.**
> >    **Response:**  Thank you for pointing this out. We have corrected it in the revised version.
> >
> > ---
> >
> > 10. **Weakness 10: Line 379: "Since" should be "since".**
> >     **Response:**  Thank you for pointing this out. We have corrected it in the revised version.
> >
> > ---
> >
> > ### **References**
> >
> > 1. Eckstein, Jonathan, and Dimitri P. Bertsekas. "On the Douglas—Rachford splitting method and the proximal point algorithm for maximal monotone operators." *Mathematical Programming* 55 (1992): 293-318.
> > 2. Davis, Damek, and Wotao Yin. "Convergence rate analysis of several splitting schemes." *Splitting methods in communication, imaging, science, and engineering* (2016): 115-163.
> > 3. Dolan, Elizabeth D., and Jorge J. Moré. "Benchmarking optimization software with performance profiles." *Mathematical Programming* 91 (2002): 201-213.
> >
> > ---
> >
> > We hope our response addresses your concerns. We sincerely appreciate your re-evaluation of our paper.

---

### Official Review · Reviewer_2QXN · 2024-11-04

**Soundness:** 3
**Presentation:** 2
**Contribution:** 2
**Rating:** 3
**Confidence:** 3

**Summary:**

In this paper, the authors consider the ergodic convergence of a preconditioned ADMM (pADMM) for solving linearly constrained convex optimization problems, which includes the Peaceman-Rachford (PR) method as a special case (where the relaxation factor $\rho =2$). Their key approach is to reformulate pADMM as an equivalent degenerate proximal point method (dPPM) and analyze its convergence using the tool of enlargements of monotone operators. Following this approach, they establish an ergodic convergence rate of $O(1/k)$ for pADMM in terms of objective error, feasibility violation, and the KKT residual. They also implemented the PR method with an ergodic sequence, incorporating an adaptive restart strategy and dynamic penalty parameter updates, for solving large-scale linear programming problems. Experimental results demonstrate improved performance over the award-winning cuPDLP solver.

**Strengths:**

- The authors analyze the ergodic convergence rate of the pADMM method with semi-proximal terms over a range of  $\rho \in (0,2]$ for the relaxation parameter. In comparison, prior studies either limit their analysis to the specific case $\rho = 1$ or primarily focus on last-iterate or asymptotic convergence.
- The authors perform extensive numerical experiments on Mittelmann's LP benchmark set and LP instances derived from MIP relaxations. The results suggest a substantial improvement over cuPDLP.

**Weaknesses:**

- My primary concern is the high similarity between this submission and the work by Chen et al. (2024), which the authors also cited in the introduction. Both papers examine pADMM-type methods with semi-proximal terms and reformulate them as a degenerate PPM method to analyze convergence. The main difference appears to be that the current submission omits the additional Halpern iteration used in Chen et al. (2024), which intuitively would simplify the analysis rather than add novel insights. Furthermore, I recommend including HPR-LP in the numerical experiments as a baseline for comparison. As it stands, the unique contribution of this paper remains unclear.
- The paper also lacks a discussion of relevant studies on the convergence of the PR method. For example, Monteiro and Sim (2018) established similar ergodic convergence rates of $O(1/k)$ for the PR method using $\epsilon$-enlargements of monotone operators. A detailed discussion of how the current result differs from prior work, such as that of Monteiro and Sim, is necessary.


----

Kaihuang Chen, Defeng Sun, Yancheng Yuan, Guojun Zhang, and Xinyuan Zhao. "HPR- LP: An implementation of an HPR method for solving linear programming." arXiv preprint arXiv:2408.12179, 2024.

Renato DC Monteiro and Chee-Khian Sim. "Complexity of the relaxed Peaceman–Rachford splitting method for the sum of two maximal strongly monotone operators." Computational Optimization and Applications 70 (2018): 763-790.

**Questions:**

- In the introduction, the authors mention that "the modified PDHG is a special case of Algorithm 1 with $\rho=1$." However, in Remark 3.1, they state that setting $\rho=1$ results in the DR method with semi-proximal terms. Could the authors clarify the difference between the DR method and the modified PDHG method in this context?
- In Assumption 2.2, it seems that the coefficient $\sigma$ is missing before $B_1^*B_1$ and $B_2^*B_2$.
- In Proposition 2.1, the authors state that $\mathcal{M}$ is an admissible preconditioner such that $(\mathcal{M}+\mathcal{T})^{-1}$ is Lipschitz continuous. However, later in Line 260, these properties are presented as an assumption. Could the authors clarify this discrepancy?

---

> ### Author Response · Authors · 2024-11-27
>
> We sincerely appreciate your positive feedback and thoughtful suggestions! To address your concerns, we provide detailed responses below.
>
> ---
>
> ### **Response to the weaknesses**
>
> 1. **Weakness 1: High similarity between this submission and the work by Chen et al. (2024). The unique contribution of this paper remains unclear.**
>    **Response:**  Thank you for your comments. While Chen et al. (2024) focus on implementing the HPR method introduced by Sun et al. (2024) for solving linear programming (LP) problems, with enhancements like adaptive restart strategies and dynamic penalty parameter adjustments, our contributions are distinct and outlined as follows:
>
>  - This paper presents significant progress in understanding the convergence properties of the PR method by proving the convergence of its ergodic sequence. More importantly, numerical results demonstrate the superior performance of the PR method’s ergodic sequence compared to that of the DR method. This finding resolves the long-standing puzzle of why the non-ergodic sequence of the PR method outperforms that of the DR method when the PR method does converge.
>    In particular, we study the convergence properties for a more general semi-proximal PR algorithm, which includes the PR method as a special case. Here, we highlight the motivation and importance of the extra semi-proximal terms in the considered semi-proximal PR algorithm: for solving large-scale convex optimization problems like LP problems, appropriate selection of the proximal operators $ \mathcal{T}_1 $ and $ \mathcal{T}_2 $ are necessary for solving the subproblems efficiently.
>
>   - We have established the $ O(1/k) $ ergodic iteration complexity for the semi-proximal PR method with respect to feasibility violation, objective error, and KKT-residual based on $ \varepsilon $-subdifferential. A key challenge arises from the fact that the concerned function may not be sub-differentiable at the ergodic point, even if it is sub-differentiable at all the points in the non-ergodic sequence. We leverage the $ \varepsilon $-subgradient to overcome this challenge.
>
> ---
>
> 2. **Weakness 2: Including HPR-LP in the numerical experiments as a baseline for comparison.**
>    **Response:**  Thank you for your comments. Compared to the numerical results presented by Chen et al. (2024), we observe that EPR-LP is slightly slower than HPR-LP, though the difference is not significant. This performance gap is understandable, given that the HPR method is a more advanced algorithm developed only recently.
>
> ---
>
> 3. **Weakness 3: More detailed comparison with Monteiro and Sim (2018).**
>
>     **Response:** Thank you for bringing this paper to our attention. Monteiro and Sim (2018) employed a hybrid proximal extra-gradient (HPE) framework to analyze the ergodic iteration complexity of the PR method, which does not prove the ergodic convergence of the PR method for solving general convex optimization problems. In contrast, we established the ergodic convergence of the semi-proximal PR method (and the PR method) based on the degenerate proximal point method (dPPM). In terms of the iteration complexity, we outline the detailed results for comparison below:
>
> - **Proposition 2.2** in our paper, the ergodic iteration complexity of dPPM:
>
>   $$
>   \mathcal{M}(w_a^k - \bar w_a^k) \in \mathcal{T}^{\bar \varepsilon_a^k}(\bar w_a^k), \quad
>   0 \leq \bar \varepsilon_a^k \leq \frac{1}{4(k+1)} \\|w^0 - w^*\\|_{\mathcal{M}}^2, \quad \forall w^* \in \mathcal{T}^{-1}(0),
>   $$
>   and
>
>   $$
>   \\|\bar w_a^k - w_a^k\\|_{\mathcal{M}} \leq \frac{1}{(k+1)} \\| w^0 - w^* \\|\_{\mathcal{M}};
>   $$
>
> - **Theorem 3.4** in Monteiro and Sim (2018), the ergodic iteration complexity of HPE
>  (Adopting the notation used in our paper, we omit $\mathcal{M}$ as $\mathcal{M} \succ 0$ in this case):
>
>   $$
>   (w^{k-1}_a - w^k_a) \in T^{\varepsilon_k^a}(\tilde{w}_k^a), \quad    0 \leq \varepsilon_k^a \leq \frac{3}{k} \left( \\|w^0 - w^*\\|^2 + \frac{\sigma^{\prime} \\|w^0 - w^*\\|^2 }{2(1-\sigma^{\prime})} \right),
>   $$
>   for any $w^* \in \mathcal{T}^{-1}(0)$ and
>
>   $$
>     \\|w^{k-1}_a - w^k_a\\|  \leq  \frac{2 \\|w^0 - w^*\\|}{k},
>   $$
>
>   where $\sigma^{\prime} < 1$ is the parameter controlling the inexactness of solving the subproblems in the HPE framework.
>
>      In our paper, we assume the operator $\mathcal{M}$ to be positive semidefinite, which allows for a broader and more general analysis. This generalization is particularly important because it enables the analysis of the ergodic iteration complexity of the semi-proximal PR method. For certain convex optimization problems, such as general LPs, appropriately chosen semi-proximal terms in the semi-proximal PR method are essential to solve the subproblem efficiently. We have included a **Remark 2.1** in the revised manuscript.

---

> ### Author Response · Authors · 2024-11-27
>
> ### **Response to the questions**
>
> 1. **Question 1: Clarify the difference between the DR method and the modified PDHG method.**
>    **Response:**  Thanks for your comments. The modified PDHG method is equivalent to the DR method (which is pADMM with $ \rho = 1 $ as described in Algorithm 1 of our manuscript) with $ \mathcal{S}_1= \lambda_1(AA^\*)I - AA^{*} $. For further details, the reviewer may refer to the works of Esser et al. (2010) and Chambolle & Pock (2011).
>
> ---
>
> 2. **Question 2: In Assumption 2.2, it seems that the coefficient $ \sigma $ is missing before $ B_1^\* B_1 $ and $ B_2^\* B_2 $.**
>    **Response:**  Thanks for your comments. We omitted the $ \sigma $ for simplicity in Assumption 2.2. For self-adjoint positive semidefinite matrices $ P $ and $ Q $, and a constant $ \sigma > 0 $, it holds that $ P + \sigma Q \succ 0 $ if and only if $ P + Q \succ 0 $. Below, we include proof for the sake of completeness:
>
>    - For $ P + Q \succ 0 \Rightarrow P + \sigma Q \succ 0 $:
>      - If $ \sigma \geq 1 $, then $ P + \sigma Q \succeq P + Q \succ 0 $.
>      - If $ \sigma < 1 $, then $ P + \sigma Q \succeq \sigma(P + Q) \succ 0 $.
>
>    - For $ P + Q \succ 0 \Leftarrow P + \sigma Q \succ 0 $:
>      - If $ \sigma \leq 1 $, then $ P + Q \succeq P + \sigma Q \succ 0 $.
>      - If $ \sigma > 1 $, then $ P + Q \succeq \frac{1}{\sigma}(P + \sigma Q) \succ 0 $.
>
> ---
>
> 3. **Question 3: Proposition 2.1 establishes that an admissible preconditioner ensures Lipschitz continuity, but Line 260 presents these properties as assumptions.**
>    **Response:**   Thank you for your comments. In Proposition 2.1, we established the Lipschitz continuity of $ (\mathcal{T} + \mathcal{M})^{-1} $ for $ \mathcal{T} $ and $ \mathcal{M} $ as defined in (6) and (7). However, the convergence results for the degenerate PPM in Theorem 2.1 hold for general $ \mathcal{T} $ and $ \mathcal{M} $.  To improve clarity, we have updated the corresponding parts in Section 2 of the revised manuscript.
>
> ---
>
> ### **References**
>
> 1. Chen, Kaihuang, Defeng Sun, Yancheng Yuan, Guojun Zhang, and Xinyuan Zhao. "HPR-LP: An implementation of an HPR method for solving linear programming." *arXiv preprint* arXiv:2408.12179 (2024).
> 2. Sun, Defeng, Yancheng Yuan, Guojun Zhang, and Xinyuan Zhao. "Accelerating preconditioned ADMM via degenerate proximal point mappings." *arXiv preprint* arXiv:2403.18618 (2024).
> 3. Monteiro, Renato DC, and Chee-Khian Sim. "Complexity of the relaxed Peaceman–Rachford splitting method for the sum of two maximal strongly monotone operators." *Computational Optimization and Applications* 70 (2018): 763-790.
> 4. Esser, Ernie, Xiaoqun Zhang, and Tony F. Chan. "A general framework for a class of first order primal-dual algorithms for convex optimization in imaging science." *SIAM Journal on Imaging Sciences* 3, no. 4 (2010): 1015-1046.
> 5. Chambolle, Antonin, and Thomas Pock. "A first-order primal-dual algorithm for convex problems with applications to imaging." *Journal of Mathematical Imaging and Vision* 40 (2011): 120-145.
>
> ---
> We hope our response addresses your concerns. We sincerely appreciate your re-evaluation of our paper.

---

### Official Review · Reviewer_cbuU · 2024-11-04

**Soundness:** 3
**Presentation:** 3
**Contribution:** 2
**Rating:** 5
**Confidence:** 3

**Summary:**

This paper studies ergodic version of PR method for solving LP based on the algorithm and its non-ergodic version of analysis in Chen et al., 2024 and Sun et al. 2024. Numerical results show the good performance over LP benchmarks.

**Strengths:**

The results appear to be sound and the paper is clearly written.

**Weaknesses:**

The overall results are somewhat expected, given the classical distinction between ergodic and non-ergodic convergence. The algorithm design and key techniques, such as the restart strategy and dynamic penalty parameter updates, were previously developed in Chen et al. (2024) and Sun et al. (2024). Additionally, the proof for the ergodic version builds on the degenerate PPM framework from Bredies et al. (2022). From a technical standpoint, achieving non-ergodic convergence is generally more challenging than ergodic convergence. The paper would be more compelling if it explored additional structural properties of LPs when using the ergodic average. Nevertheless, the numerical experiments already show some advantages when applying this approach in a straightforward manner.

**Questions:**

Although Figure 1 includes comparisons, should the non-ergodic version of the algorithm also be incorporated into Figures 2 and 3? In general comparison between ergodic and non-ergodic convergence, taking averages can compromise some desirable properties. For this specific LP task, it would be valuable to see how the computational performance for the ergodic sequence compares, providing a clearer picture of its practical implications and insights.

---

> ### Author Response · Authors · 2024-11-27
>
> ### **Response to the weaknesses**
>
> 1. **Weakness 1: Comparison with the works of Chen et al. (2024) and Sun et al. (2024).**
>    **Response:**     Thank you for your comments. We want to emphasize that the main results of this paper are different from those of Chen et al. (2024) and Sun et al. (2024). Below, we compare the differences in detail.
>
>    This paper presents significant progress in understanding the convergence properties of the PR method by proving the convergence of its ergodic sequence. More importantly, numerical results demonstrate the superior performance of the PR method’s ergodic sequence compared to that of the DR method. This finding resolves the long-standing puzzle of why the non-ergodic sequence of the PR method outperforms that of the DR method when the PR method does converge.
>
>    In particular, we study the convergence properties for a more general semi-proximal PR algorithm, which includes the PR method as a special case. Here, we highlight the motivation and importance of the extra semi-proximal terms in the considered semi-proximal PR algorithm: for solving large-scale convex optimization problems like LP problems, appropriate selection of the proximal operators $ \mathcal{T}_1 $ and $ \mathcal{T}_2 $ are necessary for solving the subproblems efficiently.
>
>    Moreover, we have established the iteration complexity guarantees of the ergodic sequence generated by the semi-proximal PR algorithm for solving the convex optimization problems. More specifically, we have established the $ O(1/k) $ ergodic iteration complexity with respect to feasibility violation, objective error, and KKT-residual based on $ \varepsilon $-subdifferential (See Theorem 2.2 in our manuscript).
>
>    In contrast, the studies by Sun et al. (2024) focused on studying the accelerating pADMM algorithm with Halpern iteration. The theoretical results contained in their paper cannot provide a convergence guarantee for semi-proximal PR or PR algorithms. The work of Chen et al. (2024) focuses on the implementation of the Halpern PR method proposed by Sun et al. (2024), which incorporates the restart strategy and dynamic penalty parameter updates.
>
>    We hope the above discussion helps to clarify the key differences between this paper and the two main references mentioned in your comments.
>
> ---
>
> 2. **Weakness 2: Achieving non-ergodic convergence is generally more challenging than ergodic convergence.**
>    **Response:**  Thank you for your comments. First, we want to mention that it is known that the non-ergodic sequence of the PR method does not converge for general convex optimization problems. This is one of the key motivations for us to study the ergodic convergence. As discussed above, we established the global ergodic convergence of the more general semi-proximal PR method in this paper.
>
>    Here, we want to highlight the challenges in the analysis of the ergodic complexity. Compared to the non-ergodic case, the analysis is more challenging. One key challenge is that the concerned function may not be sub-differentiable at the ergodic point, even if it is sub-differentiable at all the points in the non-ergodic sequence. To overcome this challenge, we leveraged the concept of an $ \varepsilon $-subgradient. More specifically, we established the ergodic $ O(1/k) $ complexity of the semi-proximal PR algorithm for solving convex optimization problems in terms of feasibility violation, objective error, and KKT residual using the $ \varepsilon $-subdifferential framework.
>    Additionally, as detailed in Appendix C in our manuscript, we establish the $ o(1/\sqrt{k}) $ non-ergodic iteration complexity of the pADMM method for $ \rho \in (0, 2) $, in terms of feasibility violation, objective error, and KKT residual. It is worthwhile mentioning that pADMM with $ \rho = 2 $ is the semi-proximal PR method.
> ---

---

> > ### Author Response · Authors · 2024-11-27
> >
> > ### **Response to the weaknesses**
> >
> > 3.**Weakness 3: The structural properties of LPs should be further explored for the ergodic iteration complexity result of the semi-proximal PR method.**
> >
> > **Response:**  Thank you for your valuable suggestions. In Appendix B of the revised manuscript, we have added the analysis of the ergodic iteration complexity of the semi-proximal PR method for solving LPs by carefully exploring the linear structures. In particular, we obtained the following results for the ergodic sequence $ \\{(\bar y^k_a, \bar z^k_a, \bar x^k_a)\\} $ generated by the semi-proximal PR method for solving LPs:
> >
> > $$
> > \begin{cases}
> >      \\|A\bar x^k_a - b\\| + \\|A^{*}\bar y^k_a + \bar z^k_a - c\\|   \leq \left( \frac{ 2 \sigma \sqrt{ \lambda_1  (A A^\*)     } + 1}{\sqrt{\sigma}} \right) \frac{R_0}{(k+1)}, \\\\
> >  -\bar z^k_a \in \partial_{\bar \varepsilon_z^k} \delta_{C}(\bar x^k_a),
> > \end{cases}
> > $$
> > corresponding to the primal infeasibility, dual infeasibility, and complementarity conditions. Here, $$ \partial_{\bar \varepsilon_z^k} \delta_{C}(\bar x^k_a) = \\{ z \in \mathbb{R}^{n} \mid \langle z, x^{\prime} - \bar x^k_a \rangle \leq \bar{\varepsilon}_z^k, \forall x^{\prime} \in C \\},$$
> >
> > $\bar \varepsilon_z^k \leq \frac{1}{4(k+1)} R_{0}^2 $, and $ R_0 $ represents the weighted distance between the initial point and the solution set.
> >
> > ### **Response to the questions**
> >
> > 1. **Question 1: Although Figure 1 includes comparisons, should the non-ergodic version of the algorithm also be incorporated into Figures 2 and 3?**
> >    **Response:**  Thank you for your question. The purpose of Figure 1 in the manuscript is to demonstrate the non-convergence of the non-ergodic sequence generated by the PR method for solving LPs. This comparison highlights the superior performance of the ergodic sequence of the PR method over its non-ergodic sequence. Due to the limited time in the discussion period, we decided not to include the non-ergodic PR in the comparison.
> >
> > 2. **Question 2: In general comparison between ergodic and non-ergodic convergence, taking averages can compromise some desirable properties.**
> >    **Response:**  Thank you for your insightful comment. We agree that taking averages can compromise some desirable properties, such as the solution sparsity. Based on our numerical experience, the restart strategy in the EPR-LP algorithm can alleviate this issue caused by the ergodic sequence.
> >
> >    As the non-ergodic sequence of the PR method may fail to converge for solving LPs, we compared the EPR-LP solver with the non-ergodic sequence of the pADMM algorithm ($\rho = 1.6$). When solving the optimal transport problem, EPR-LP equipped with a restart strategy requires only about 5,000 iterations to achieve a solution with the same sparsity as the non-ergodic sequence of pADMM. In terms of solving time and iteration number, the non-ergodic pADMM requires 8,529,300 iterations and 19,514.3 seconds to achieve a solution with a tolerance of $10^{-8}$, whereas EPR-LP achieves the same result in just 220,050 iterations and 514.9 seconds. For more details, please refer to Figure 4 in Appendix G of the revised version.
> >
> >    To sum up, while taking averages may introduce certain compromises, the significant computational advantages of the ergodic sequence, especially when equipped with restart strategies, make it a highly appealing choice.
> >
> > ---
> >
> > ### **Reference**
> >
> > 1. Chen, Kaihuang, Defeng Sun, Yancheng Yuan, Guojun Zhang, and Xinyuan Zhao. "HPR-LP: An implementation of an HPR method for solving linear programming." *arXiv preprint* arXiv:2408.12179 (2024).
> > 2. Sun, Defeng, Yancheng Yuan, Guojun Zhang, and Xinyuan Zhao. "Accelerating preconditioned ADMM via degenerate proximal point mappings." *arXiv preprint* arXiv:2403.18618 (2024).
> >
> > ---
> >
> > We hope that our response resolves your concerns. We are happy to offer further explanations if needed and would greatly appreciate it if you could kindly re-evaluate our paper. Thank you.

---

### Note · Authors · 2025-01-12

I have read and agree with the venue's withdrawal policy on behalf of myself and my co-authors.